# Endothelial EphB4 maintains vascular integrity and transport function in adult heart

Guillermo Luxán[1†], Jonas Stewen[1†], Noelia Díaz[2], Katsuhiro Kato[1], Sathish K Maney[1], Anusha Aravamudhan[1], Frank Berkenfeld[1], Nina Nagelmann[3], Hannes CA Drexler[4], Dagmar Zeuschner[5], Cornelius Faber[3], Hermann Schillers[6], Sven Hermann[7], John Wiseman[8], Juan M Vaquerizas[2], Mara E Pitulescu[1‡]*, Ralf H Adams[1,9‡]*

[1]Department of Tissue Morphogenesis, Max Planck Institute for Molecular Biomedicine, Münster, Germany; [2]Regulatory Genomics Laboratory, Max Planck Institute for Molecular Biomedicine, Münster, Germany; [3]Department of Clinical Radiology, University Hospital Münster, Münster, Germany; [4]Bioanalytical Mass Spectrometry Unit, Max Planck Institute for Molecular Biomedicine, Münster, Germany; [5]Electron Microscopy Unit, Max Planck Institute for Molecular Biomedicine, Münster, Germany; [6]Institute for Physiology II, University of Münster, Münster, Germany; [7]European Institute for Molecular Imaging, University of Münster, Münster, Germany; [8]Discovery Biology, Discovery Sciences, IMED Biotech Unit, AstraZeneca, Gothenburg, Sweden; [9]Faculty of Medicine, University of Münster, Münster, Germany

*For correspondence:
mara.pitulescu@mpi-muenster.
mpg.de (MEP);
ralf.adams@mpi-muenster.mpg.
de (RHA)

[†]These authors contributed equally to this work
[‡]These authors also contributed equally to this work

**Abstract** The homeostasis of heart and other organs relies on the appropriate provision of nutrients and functional specialization of the local vasculature. Here, we have used mouse genetics, imaging and cell biology approaches to investigate how homeostasis in the adult heart is controlled by endothelial EphB4 and its ligand ephrin-B2, which are known regulators of vascular morphogenesis and arteriovenous differentiation during development. We show that inducible and endothelial cell-specific inactivation of *Ephb4* in adult mice is compatible with survival, but leads to rupturing of cardiac capillaries, cardiomyocyte hypertrophy, and pathological cardiac remodeling. In contrast, EphB4 is not required for integrity and homeostasis of capillaries in skeletal muscle. Our analysis of mutant mice and cultured endothelial cells shows that EphB4 controls the function of caveolae, cell-cell adhesion under mechanical stress and lipid transport. We propose that EphB4 maintains critical functional properties of the adult cardiac vasculature and thereby prevents dilated cardiomyopathy-like defects.

## Introduction

Dilated cardiomyopathy (DCM) is a common and irreversible type of heart disease. It is the third most common case of heart failure and the most frequent cause for heart transplantation (*American Heart Association et al., 2006*) with an estimated prevalence of 40 in 100.000 people (*Manolio et al., 1992*). Up to 80% of DCM patients present heart failure symptoms (*Dec and Fuster, 1994*). Frequently, the disease first affects the left ventricle, where the muscle starts to remodel, leading to increased end-diastolic and end-systolic volumes (*Mestroni et al., 2014*).

The heart is a highly vascularized organ and capillaries reside in close proximity to almost every cardiomyocyte with a cellular ratio between cardiomyocytes and endothelial cells (ECs) of 1:3

(*Brutsaert, 2003*; *Hsieh et al., 2006*). Around 35% of the known cases of DCM are due to mutations in genes that mainly encode myocardial cytoskeletal, sarcomeric and nuclear envelope proteins, but there are also acquired causes that include metabolic and endocrine disruptions (*Garfinkel et al., 2018*; *Weintraub et al., 2017*). Yet, most of the DCM cases are considered idiopathic, as the underlying cause is unknown (*Towbin et al., 2006*). Capillary ECs constitute a functional interface between the circulation and the myocardium. Their proximity to cardiomyocytes makes ECs ideally suited to control cardiac muscle cell homeostasis by, for example, the regulation of nutrient delivery and organ metabolism. Fatty acids are a crucial metabolic substrate for the heart (*van der Vusse, 2000*) and alterations in cardiomyocyte nutrient preference, such as enhanced glucose uptake and glycolysis, have been observed in multiple pathological conditions (*Ritterhoff and Tian, 2017*). Highlighting the importance of ECs in cardiac homeostasis, endothelial mutations in the laminin α4 subunit and integrin-linked kinase can cause cardiomyopathy (*Knöll et al., 2007*) and disruption of endothelial Notch signaling impairs fatty acid transport and leads to pathological heart remodeling (*Jabs et al., 2018*).

The receptor tyrosine kinase EphB4 and its ligand, the transmembrane protein ephrin-B2, regulate critical aspects of EC behavior in a cell-cell contact-dependent fashion. Signaling by ephrin-B2 and EphB4 has been implicated in the regulation of sprouting angiogenesis, vascular morphogenesis, arteriovenous differentiation (*Pitulescu and Adams, 2010*), and cancer (*Pasquale, 2010*), but little is known about the function of this ligand-receptor pair in adult organ homeostasis. Here, we report that inducible inactivation of the *Ephb4* gene in the adult endothelium causes a cardiac phenotype that resembles key features of DCM. Mutant ECs are prone to break upon mechanical stress and are not able to transport fatty acids resulting in cardiomyocyte hypertrophy and heart remodeling similar to defects observed in dilated cardiomyopathy. Our results identify ephrin-B2 and EphB4 as critical regulators of the cardiac vasculature and heart homeostasis.

## Results

### Loss of endothelial EphB4 causes heart hypertrophy

The receptor tyrosine kinase EphB4 and its ligand ephrin-B2 are expressed in the capillary plexus of the adult coronary vasculature. In addition, EphB4 is expressed in large veins, whereas ephrin-B2 is restricted to arteries (*Figure 1—figure supplement 1A,B*). To determine the role of signaling by EphB4 and ephrin-B2 in the adult cardiac endothelium, we bred mice bearing a conditional *Ephb4* loss-of-function allele (*Wang et al., 2015*) with $Cdh5^{CreERT2}$ (*Wang et al., 2010*) transgenic animals expressing tamoxifen-inducible Cre recombinase specially in ECs. Following the injection of adult mice at 8 weeks of age with 4-hydroxy tamoxifen, hearts were analyzed 4 weeks afterwards. Cre activity was monitored using *R26-mTmG* Cre reporter mice (*Muzumdar et al., 2007*), in which all cells express membrane-anchored Tomato fluorescent protein but switch to the expression of membrane-associated green fluorescent protein (GFP) in a Cre-controlled fashion. This approach revealed a very high recombination efficiency in cardiac ECs at 12 weeks (*Figure 1—figure supplement 1C*). Moreover, EphB4 expression was strongly diminished in the coronaries of $Cdh5^{CreERT2}$ $Ephb4^{flox/flox}$ ($Ephb4^{\Delta EC}$) hearts when compared to Cre-negative littermate controls (*Figure 1—figure supplement 1D*). This result was further confirmed by the drastic reduction of EphB4 protein in $Ephb4^{\Delta EC}$ whole heart lysates analyzed by Western blot (*Figure 1—figure supplement 1E*).

Morphometric characterization revealed that heart size and weight were increased in $Ephb4^{\Delta EC}$ mutants. On average, mutant hearts weighed 20% more than controls, whereas total body weight was not significantly altered. Normalization of heart weight to tibia length (*Yin et al., 1982*) confirmed this finding (*Figure 1A*). Histologic analysis of the left ventricle's cardiac wall revealed a significant enlargement of the cardiomyocytes in $Ephb4^{\Delta EC}$ hearts (*Figure 1B*), indicating hypertrophic remodeling of the heart muscle. Moreover, echocardiography analysis of mutant hearts showed a reduced heart ejection fraction and larger left ventricle diastolic and systolic volumes (*Figure 1C* and *Figure 1—videos 1* and *2*). Cardiac magnetic resonance imaging (CMRI) revealed that the walls of the organ were significantly thinner at the interventricular septum and that the mass of the left ventricle was lower, whereas the right ventricle was not affected at this stage (*Figure 1D* and *Figure 1—videos 3–6*). These observations indicate ongoing dilation of $Ephb4^{\Delta EC}$ mutant ventricles.

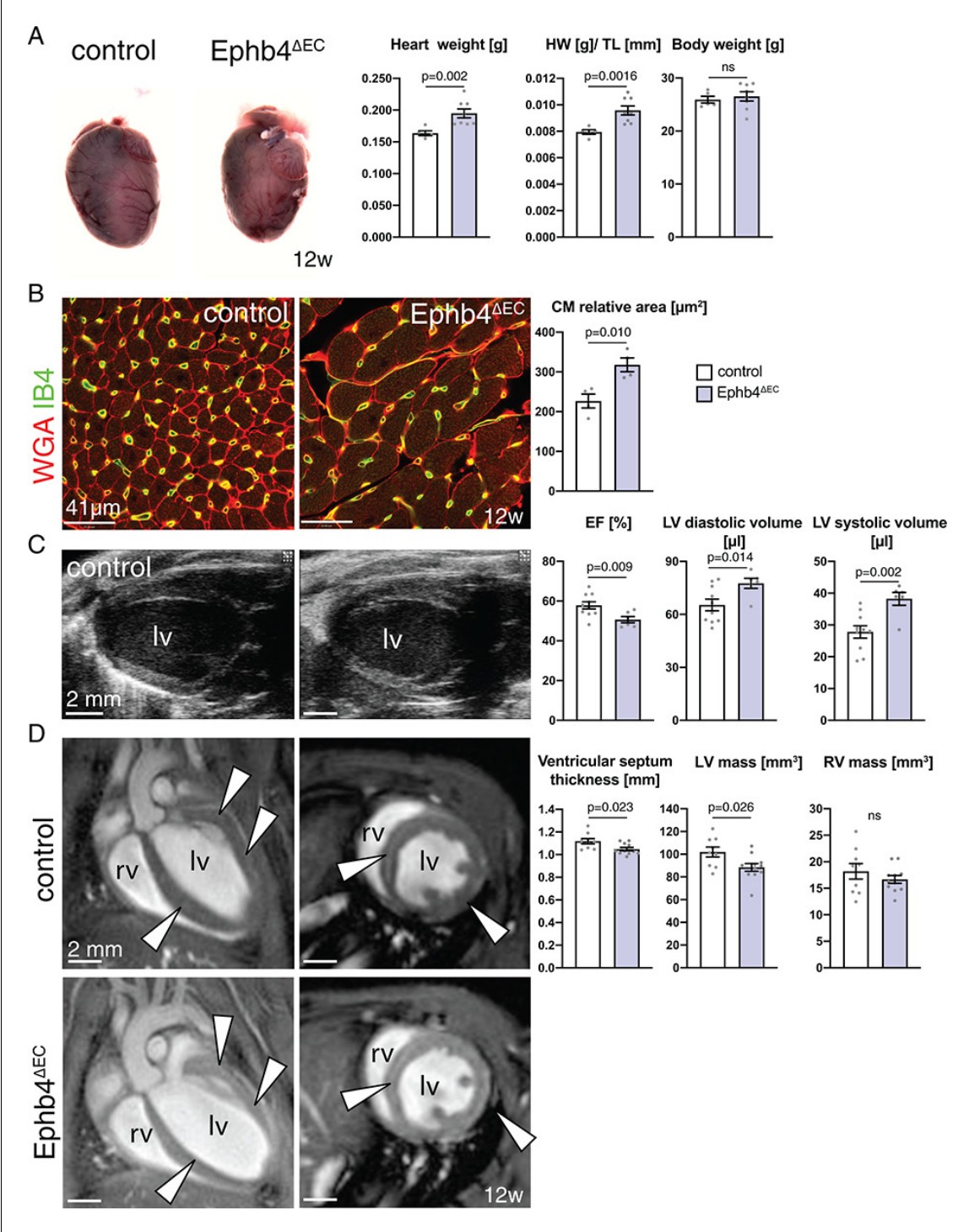

**Figure 1.** Heart defects in adult *Ephb4* mutants. (**A**) Freshly dissected Ephb4$^{\Delta EC}$ and littermate control hearts at 12 weeks of age. Heart weight and heart weight/tibia length index (HW/TL) ratio are increased in Ephb4$^{\Delta EC}$ mutants, whereas body weight remains unchanged. N = 5 for control and N = 8 for Ephb4$^{\Delta EC}$. (**B**) Immunohistochemistry on cross sections of control and Ephb4$^{\Delta EC}$ hearts at 12 weeks. Panels show the inner part of the left ventricular wall with significantly increased cardiomyocyte relative area in Ephb4$^{\Delta EC}$ samples. N = 4 for both genotypes. (**C**) Echocardiography analysis of 12 week-old animals. Ejection fraction (EF) is significantly reduced in mutant mice while left ventricle diastolic and systolic volumes are increased. lv, left ventricle. N = 10 for control and N = 6 for Ephb4$^{\Delta EC}$ (**D**) CMRI sections of hearts showing four-chamber (left) and short axis (right) views of control and Ephb4$^{\Delta EC}$ at 12 weeks of age. Arrowheads indicate the ventricular septum and the wall of the left ventricle. rv, right ventricle; lv, left ventricle. Ventricular septum thickness and left ventricular (lv) mass are significantly reduced in Ephb4$^{\Delta EC}$ mice, whereas right ventricular (rv) mass remains

*Figure 1 continued on next page*

*Figure 1 continued*

unchanged. N = 9 for control and N = 11 for Ephb4$^{\Delta EC}$. Data represented as mean ± s.e.m. *P* values calculated by unpaired two-tailed *t* test with Welch's correction. ns, not significant.

The online version of this article includes the following video, source data, and figure supplement(s) for figure 1:

**Source data 1.** Source data for *Figure 1A,B,C,D*.
**Figure supplement 1.** Inactivation of *Ephb4* in adult ECs.
**Figure 1—video 1.** Left ventricle long axis view.
https://elifesciences.org/articles/45863#fig1video1
**Figure 1—video 2.** Left ventricle long axis view.
https://elifesciences.org/articles/45863#fig1video2
**Figure 1—video 3.** Four chamber view.
https://elifesciences.org/articles/45863#fig1video3
**Figure 1—video 4.** Short axis view.
https://elifesciences.org/articles/45863#fig1video4
**Figure 1—video 5.** Four chamber view.
https://elifesciences.org/articles/45863#fig1video5
**Figure 1—video 6.** CMRI 12 week-old Ephb4$^{\Delta EC}$ mutant.
https://elifesciences.org/articles/45863#fig1video6

## Loss of EphB4 affects the stability of the coronary plexus

Next, we examined the cell-autonomous effects of *Ephb4* inactivation in coronary vessels. Interestingly, histologic analysis revealed that mutant mice have reduced vascular density with less coronary vessel branch points (*Figure 2A*), an effect that could be partially influenced by the increased cardiomyocyte area. In addition, immunostaining against PDGFRβ, a marker of pericytes and vascular smooth muscle cells associated with the endothelial monolayer, showed a significant loss of mural cell coverage in the blood vessels of the ventricle, which was especially prominent in certain capillaries (*Figure 2B*). Moreover, extravasated erythrocytes were detected in the mutant myocardium, indicating the presence of microhemorrhages in the Ephb4$^{\Delta EC}$ ventricular wall (*Figure 2C*). Nevertheless, mutant ECs did not show increased cell death, as indicated by comparable TUNEL staining in control and mutant ventricles (*Figure 2—figure supplement 1A*). Analysis of tissue oxygenation through Pimonidazole administration did not indicate appreciable tissue hypoxia, arguing for sufficient oxygen transport through the Ephb4$^{\Delta EC}$ endothelium (*Figure 2—figure supplement 1B,C*). It is also noteworthy that loss of EphB4 in ECs did not induce fibrosis in the mutant myocardium (*Figure 2—figure supplement 2A,B*).

To study the phenotype of Ephb4$^{\Delta EC}$ ECs in more detail, we analyzed the vasculature of the left ventricular wall by transmission electron microscopy (TEM). High magnification images revealed that mutant ECs were irregularly arranged and displayed an expanded cytoplasm largely devoid of vesicles, whereas caveolar structures were abundant along the basolateral membrane (*Figure 2D* and *Figure 2—figure supplement 3A–C*). Moreover, mutant cardiomyocytes showed the accumulation of glycogen granules in their cytoplasm close to the mitochondria (*Figure 2—figure supplement 3D*). These observations suggest that the transport through the endothelium and cardiomyocyte metabolism might be affected in Ephb4$^{\Delta EC}$ mutant hearts. TEM also revealed ruptures of the capillaries explaining the microhemorrhages mentioned above (*Figure 2D*).

## Inactivation of the gene encoding ephrin-B2 in adult ECs

To address the role of ephrin-B2 in EphB4-mediated signaling, we interbred mice bearing a conditional *Efnb2* loss-of-function allele, *Efnb2$^{flox/flox}$* (*Grunwald et al., 2004*) with *Cdh5$^{CreERT2}$* transgenic mice and treated *Cdh5$^{CreERT2}$ Efnb2$^{flox/flox}$* (Efnb2$^{\Delta EC}$) mice and Cre-negative littermate controls with 4-hydroxy tamoxifen in the same way as described above for Ephb4$^{\Delta EC}$ mutants. The phenotype observed upon EC-specific loss of ephrin-B2 was milder than the defects in Ephb4$^{\Delta EC}$ hearts. While Efnb2$^{\Delta EC}$ heart weight was not altered (*Figure 3A*), the cardiomyocyte relative area was significantly increased (*Figure 3B*). Vascular density (*Figure 3C*) and perivascular coverage of the capillaries (*Figure 3D*) was not appreciably altered. Interestingly, the changes observed in Efnb2$^{\Delta EC}$ mutants were very similar to the ones seen in Ephb4$^{\Delta EC}$ hearts at 10 weeks of age, that is 2 weeks after

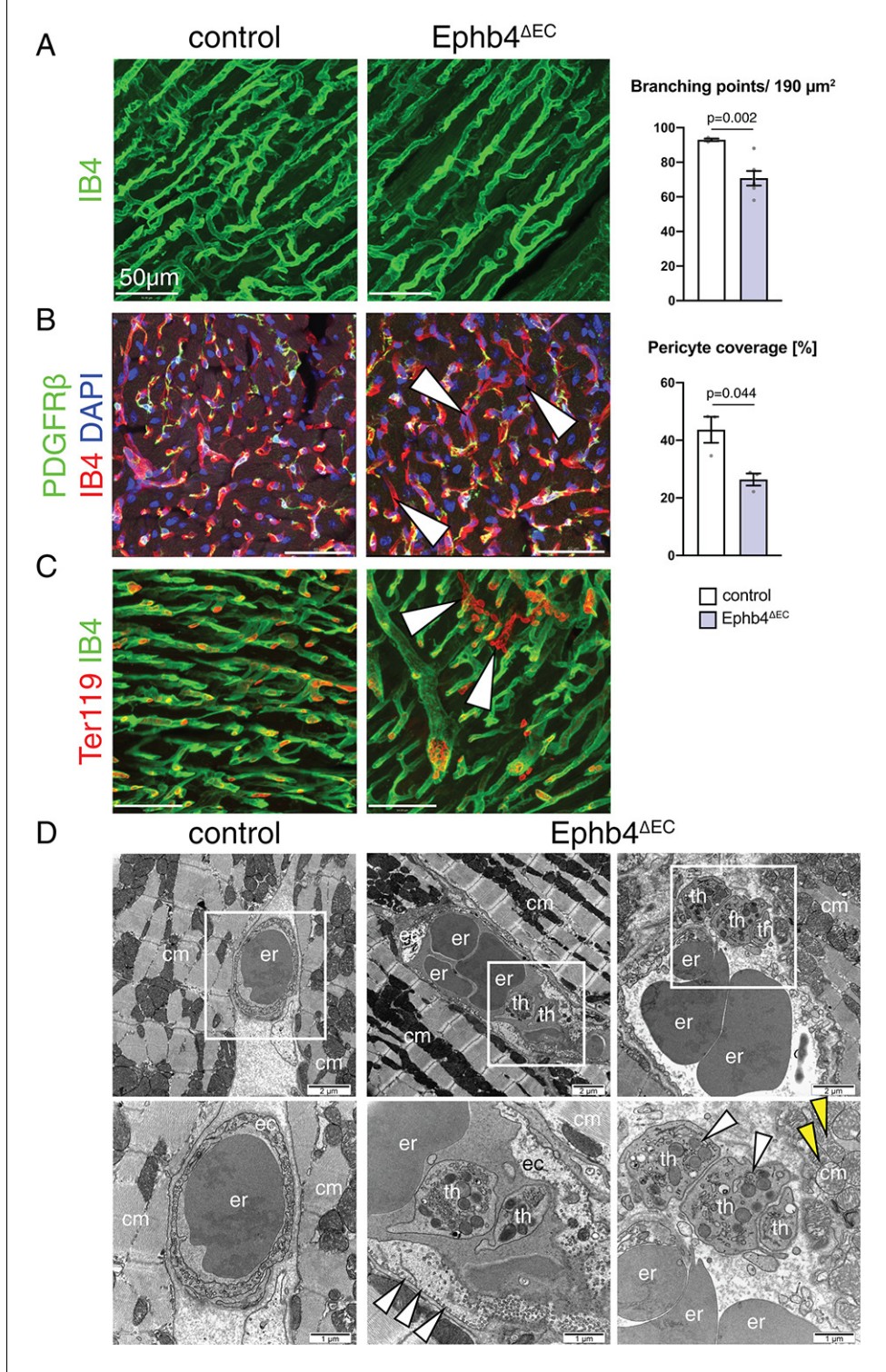

**Figure 2.** *Ephb4* inactivation compromises cardiac vascular integrity. (A–C) Immunostaining of cross sections through 12 week-old control and Ephb4^ΔEC hearts. Confocal images show the outer (A, C) and inner part (B) of the left ventricle. (A) Vascular density, measured by number of branching points, is significantly reduced in Ephb4^ΔEC samples. N = 3 for control and N = 6 for Ephb4^ΔEC. (B) Pericyte coverage is reduced in Ephb4^ΔEC hearts (arrowheads mark affected capillaries). N = 3 per genotype. (C) Presence of microhemorrhages (arrowheads mark erythrocytes in the mutant myocardium). (D) Electron micrographs of control and Ephb4^ΔEC capillaries. Bottom images are higher magnifications of boxed areas in upper panels. White arrowheads indicate accumulation of caveolar vesicles at the mutant endothelial basolateral membrane (center) and thrombocytes in a vascular

*Figure 2 continued on next page*

*Figure 2 continued*

rupture (right). Yellow arrowheads indicate mitochondrial glycogen accumulation. Erythrocytes (er), thrombocytes (th), cardiomyocytes (cm) and endothelial cells (ec) are indicated. Data represented as mean ± s.e.m. *P* values calculated by unpaired two-tailed *t* test with Welch's correction.

The online version of this article includes the following source data and figure supplement(s) for figure 2:

**Source data 1.** Source data for *Figure 2A,B*.
**Figure supplement 1.** Analysis of cell death and hypoxia.
**Figure supplement 2.** *Ephb4* inactivation does not induce fibrosis in heart.
**Figure supplement 3.** Ultrastructural analysis of *Ephb4* mutant hearts.
**Figure supplement 3—source data 1.** Source data for *Figure 2—figure supplement 3C*.

tamoxifen induction. Ephb4$^{\Delta EC}$ hearts at this stage did not show yet a significant weight increase (*Figure 3—figure supplement 1A*) but their cardiomyocytes were already hypertrophic, resulting in a significant increase in relative area (*Figure 3—figure supplement 1B*). Ephb4$^{\Delta EC}$ vascular density (*Figure 3—figure supplement 1C*) and perivascular coverage of the capillaries (*Figure 3—figure supplement 1D*) was not yet affected at 10 weeks. Both, Efnb2$^{\Delta EC}$ hearts at 12 weeks and Ephb4$^{\Delta EC}$ hearts at 10 weeks hearts presented microhemorrhages (*Figure 3—figure supplement 2A,B*).

## Heart-specific muscle cell remodeling after loss of EphB4 in ECs

To investigate whether the observed muscle cell defects are confined to heart, we analyzed skeletal muscle, a tissue where EphB4 and ephrin-B2 are expressed in the vasculature in a similar pattern as in heart (*Figure 4A*). In contrast to the defects seen in cardiac muscle, skeletal muscle was unaffected after the EC-specific inactivation of *Ephb4*. Both vascular density (*Figure 4B*) and pericyte coverage (*Figure 4C*) of capillaries remained unchanged and, most importantly, the relative area occupied by myocytes was not significantly altered (*Figure 4D*). Likewise, the vasculature of other organs, namely kidney and liver, was not appreciably altered in EphB4$^{\Delta EC}$ mutants (*Figure 4—figure supplement 1A–C*).

EphB4 mutations are known to cause *hydrops fetalis* (*Martin-Almedina et al., 2016*) due to the role of EphB4 in the development of lympho-venous valves (*Zhang et al., 2015*). In order to rule out that the Ephb4$^{\Delta EC}$ phenotype is caused by defects in lymphatic vessels, which are also targeted by Cdh5$^{CreERT2}$ transgenic allele, we interbred *Ephb4* conditional mice with Prox1$^{CreERT2}$ animals (*Bazigou et al., 2011*). Despite efficient EphB4 deletion in lymphatic vessels (Ephb4$^{\Delta LEC}$) after 4-hydroxy tamoxifen administration (*Figure 4—figure supplement 2A*), heart size and cardiomyocyte relative area remained unaffected (*Figure 4—figure supplement 2B,C*). These results argue that the heart phenotype arises due to loss of EphB4 in blood vessels but not the lymphatic vasculature.

## Gene expression analysis indicates EphB4-dependent metabolic defects

To gain insight into the molecular processes occurring in Ephb4$^{\Delta EC}$ mutant hearts, we performed global gene expression analysis of mutant and littermate control ventricles at 12 weeks of age by RNA sequencing. Differential gene expression analysis identified 529 genes (p<0.05) of which 268 were upregulated and 261 downregulated (*Figure 5A*; *Figure 5—figure supplement 1A* and *Figure 5—source data 1*). Gene ontology (GO) classification of differentially expressed genes was calculated using the database Enrichr (http://amp.pharm.mssm.edu/Enrichr/) (*Chen et al., 2013*; *Kuleshov et al., 2016*) (*Figure 5B–F*). First, the analysis revealed that genes related to human disease, and in particular to coronary artery disease, hypertrophic cardiomyopathy (HCM) and DCM were dysregulated in Ephb4$^{\Delta EC}$ hearts (*Figure 5B*). Further GO analysis of biological processes revealed that genes related to the regulation of carbohydrate metabolism and the positive regulation of glycolysis were upregulated in mutant samples. These observations are interesting because glucose consumption and glycolysis in the heart are indicators of cardiomyopathy (*Ritterhoff and Tian, 2017*) (*Figure 5C* and *Figure 5—source data 1*). We also observed that genes related to extracellular matrix (ECM) organization were downregulated in Ephb4$^{\Delta EC}$ hearts (*Figure 5D* and *Figure 5—source data 1*). Pathway analysis using the Kyoto Encyclopedia of Genes and Genomes (KEGG) returned several interesting pathways enriched in affected genes. In particular, two signaling pathways strongly related to cardiac metabolism and disease were altered: FoxO signaling and the genes *Agap2*, *Pten*, *Igf1*, *Pik3r1*, *Foxo3* and *Pck2* were upregulated, whereas mTOR signaling and

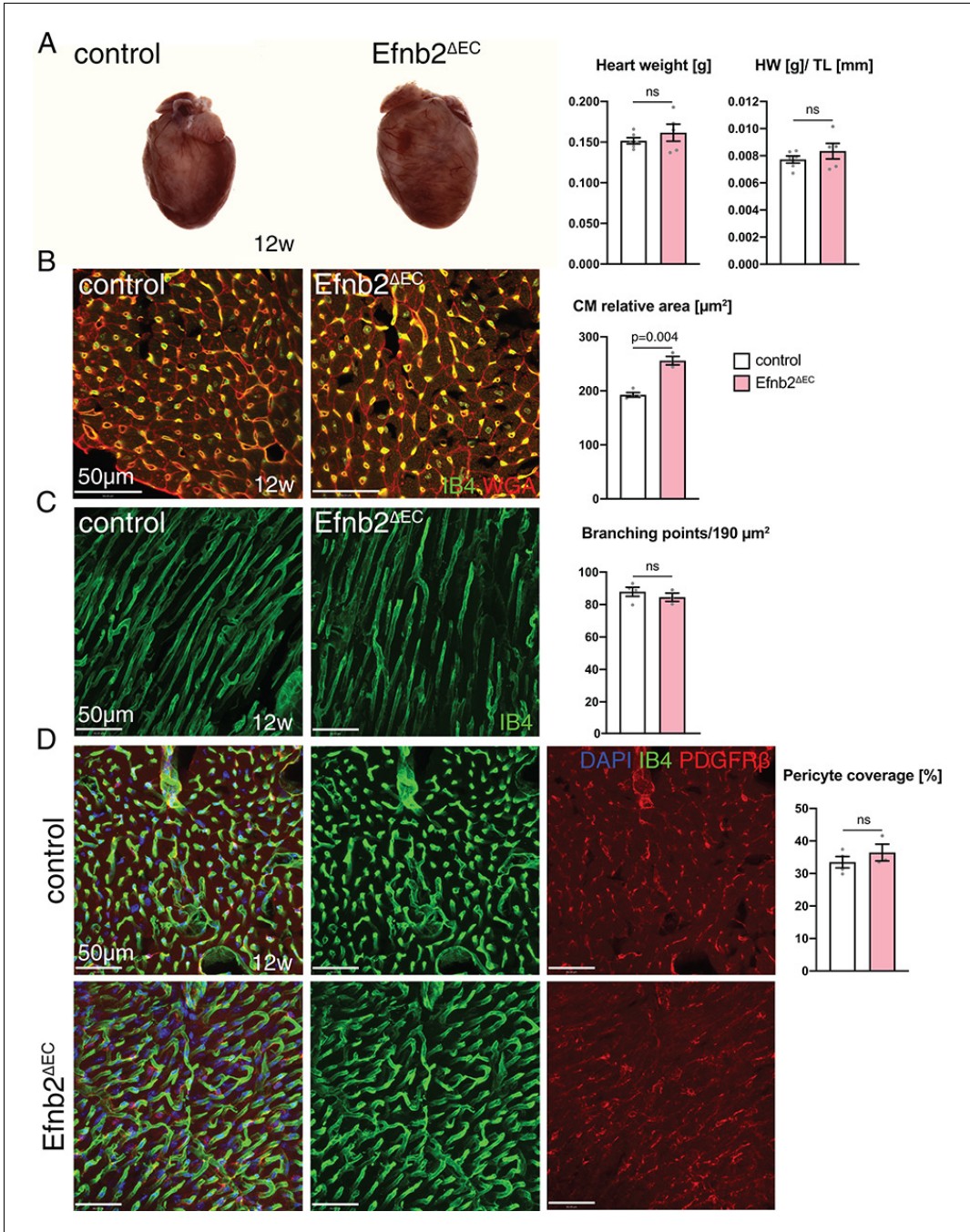

**Figure 3.** Heart phenotype after *Efnb2* inactivation in the vascular endothelium. (**A**) Efnb2ᴬᴱᶜ and littermate control hearts at 12 weeks of age. Heart weight and heart weight/tibia length index (HW/TL) remain unaffected in Efnb2ᴬᴱᶜ mutants. N = 6 for control and N = 5 for Efnb2ᴬᴱᶜ. (**B–D**) Immunohistochemistry on sections of control and Efnb2ᴬᴱᶜ hearts. Panels show the inner (**B, D**) and the outer part (**C**) of the wall of the left ventricle. (**B**) Cardiomyocyte (CM) relative area is significantly increased in Efnb2ᴬᴱᶜ hearts. N = 4 for control and N = 3 for Efnb2ᴬᴱᶜ. (**C**) Vascular density, measured by number of branching points, remains unchanged in Efnb2ᴬᴱᶜ samples. N = 4 for control and N = 3 for Efnb2ᴬᴱᶜ. (**D**) Cardiac pericyte coverage is not reduced in Efnb2ᴬᴱᶜ mutants. N = 4 for control and N = 3 for Efnb2ᴬᴱᶜ. Data represented as mean ± s.e.m. *P* values calculated by unpaired two-tailed *t* test with Welch's correction. ns, not significant.

The online version of this article includes the following source data and figure supplement(s) for figure 3:

**Source data 1.** Source data for *Figure 3A,B,C,D*.
**Figure supplement 1.** Cardiomyocyte hypertrophy at 2 weeks after *Ephb4* inactivation.
**Figure supplement 1—source data 1.** Source data for *Figure 3—figure supplement 1A,B,C,D*.
**Figure supplement 2.** Microhemorrhages in *Ephb4* and *Efnb2* mutant hearts.

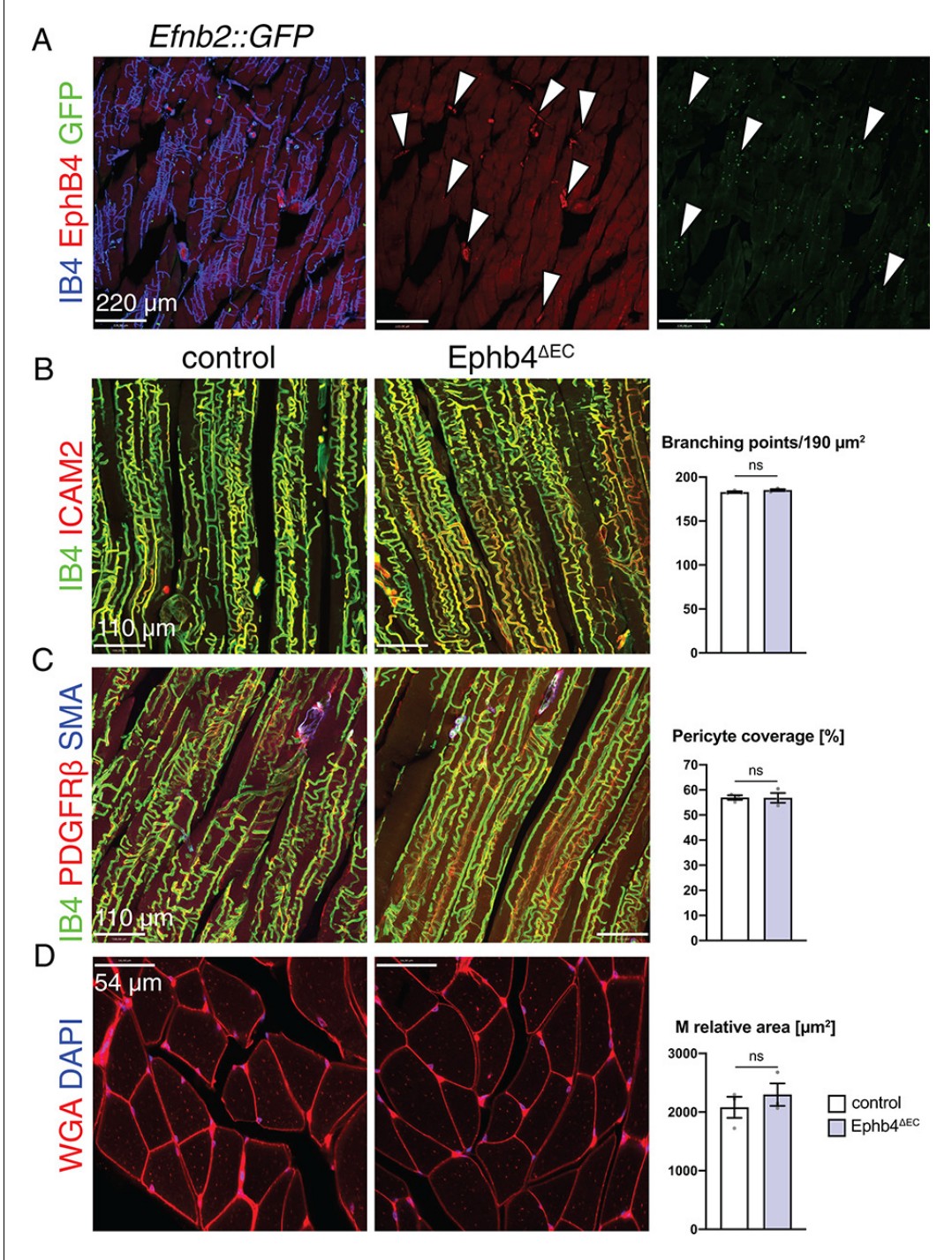

**Figure 4.** EphB4 inactivation does not affect the skeletal muscle. (A–D) Immunohistochemistry on longitudinal sections of control and Ephb4$^{\Delta EC}$ gastrocnemius at 12 weeks of age. (A) EphB4 and ephrin-B2 (GFP in *Efnb2::GFP* reporter line) are expressed in capillaries (arrowheads). (B) Vascular density, measured by number of branching points, is not changed in Ephb4$^{\Delta EC}$ gastrocnemius. (C) Pericyte coverage and (D) myocyte (M) relative area is not reduced in Ephb4$^{\Delta EC}$ gastrocnemius. N = 3 for both genotypes. Data represented as mean ± s.e.m. *P* values calculated by unpaired two-tailed *t* test with Welch's correction. ns, not significant.

The online version of this article includes the following source data and figure supplement(s) for figure 4:

**Source data 1.** Source data for *Figure 4B,C,D*.
**Figure supplement 1.** Liver and kidney vasculature after *Ephb4* inactivation.
**Figure supplement 2.** Lymphatic inactivation of *Ephb4* in adults does not produce cardiac defects.

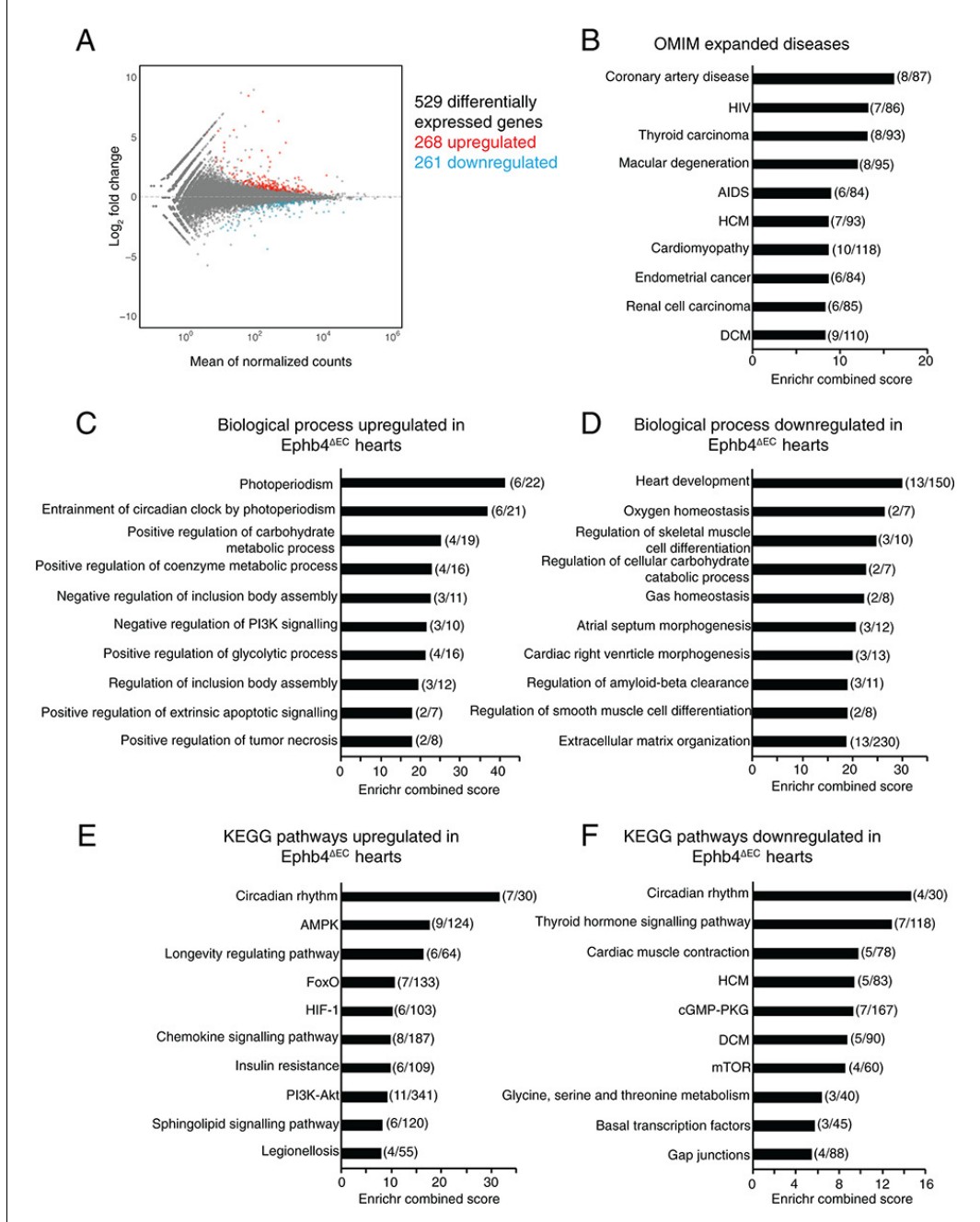

**Figure 5.** RNA-seq analysis of total heart suggests a metabolic shift in *Ephb4* mutants. (A–F) Global gene expression analysis of Ephb4$^{\Delta EC}$ ventricles by RNA sequencing at 12 weeks. (A) MA plot representing the 529 differentially expressed genes (p<0.05) including 268 upregulated genes (red) and 261 downregulated genes (blue). (B–F) Representation of the ten most significant functional categories in each group revealed by gene ontology analysis using the Enrichr data base. Graphs represent Enrichr combined score that combines *P* value and Z score. Numbers in brackets represent the number of differentially expressed genes in the corresponding category. (B) Human disease enriched terms according to the Online Mendelian Inheritance in Man (OMIM). Biological processes upregulated (C) and downregulated (D) in Ephb4$^{\Delta EC}$ ventricles. Kyoto Encyclopedia of Genes and Genomes (KEGG) pathways upregulated (E) and downregulated (F) in Ephb4$^{\Delta EC}$ ventricles. HCM, hypertrophic cardiomyopathy; DCM, dilated cardiomyopathy.
The online version of this article includes the following source data and figure supplement(s) for figure 5:

**Source data 1.** RNA-seq analysis of Ephb4$^{\Delta EC}$ and control mouse heart ventricles.
**Source data 2.** Global proteome analysis of Ephb4$^{\Delta EC}$ and control mouse heart ventricles.
**Figure supplement 1.** Genome and Proteome analysis of adult *Ephb4* mutant ventricles.
**Figure supplement 1—source data 1.** Source data for *Figure 5—figure supplement 1B,C*.

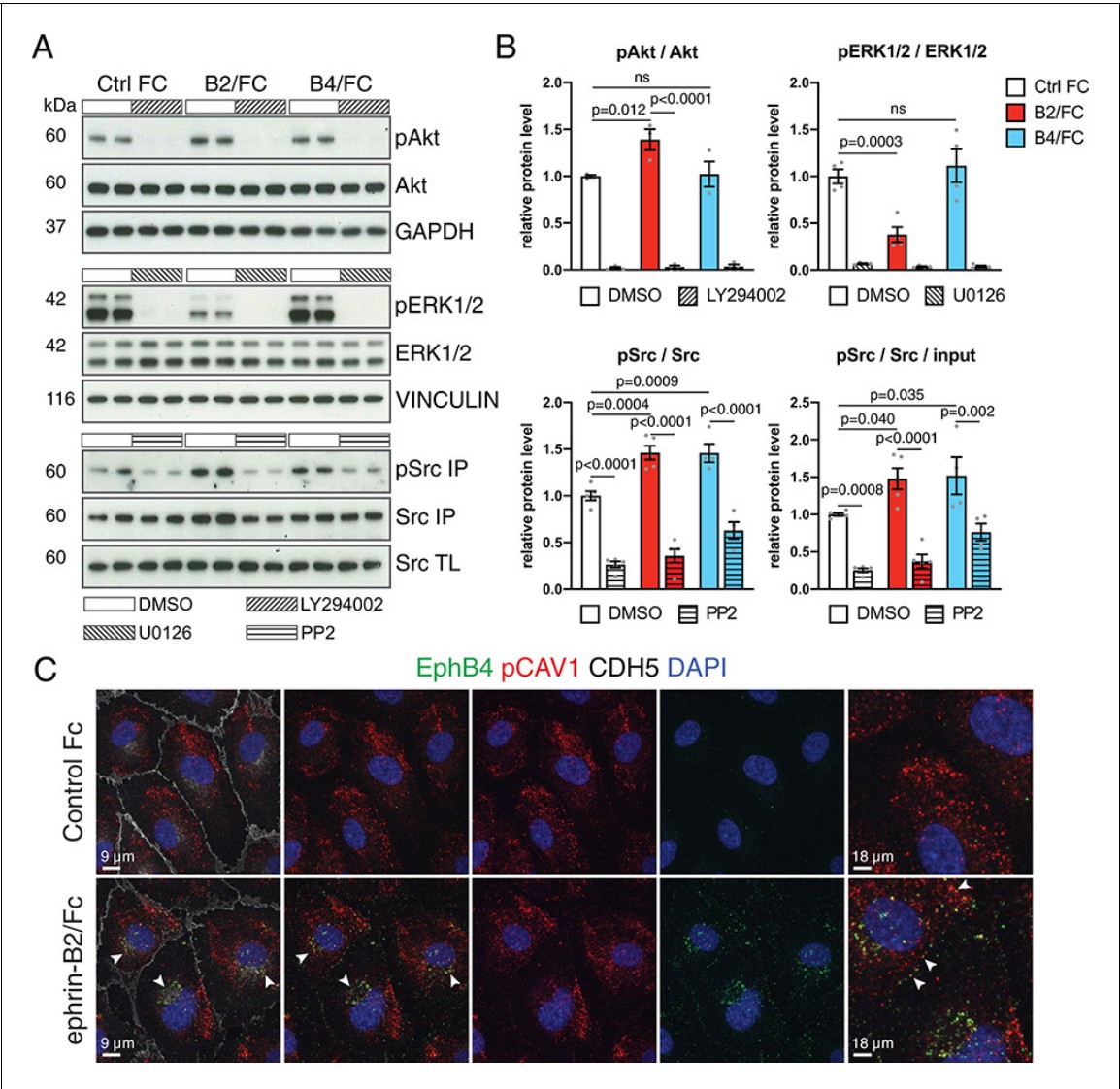

**Figure 6.** EphB4-induced signaling in cultured ECs. (**A**) Western blot analysis of lysates from cultured HUVECs stimulated with control human IgG/Fc (Ctrl Fc), human ephrin-B2/Fc (B2/Fc) or mouse EphB4/Fc (B4/Fc) (4 µg/ml, preclustered with 10µg/ml goat anti-human IgG), concurrently treated with inhibitors (LY294002, U0126, PP2) or DMSO for 30 min. Active Src (pSrc) was determined by anti-Src immunoprecipitation (IP) followed by anti-Src or anti-phosphotyrosine (4G10) immunoblotting. Bottom panel indicates Src input in total cell lysate (TL). Molecular weights (kDa) are indicated. N = 3 for all treatments. (**B**) Graphs show relative quantitation of pAkt/Akt, pERK1/2/ERK1/2, pSrc/Src and pSrc/Src/input. N = 3 for LY294002 and U0126 experiments. For PP2 experiment N = 4 for Ctrl Fc/DMSO, B4/Fc/DMSO and B4/Fc/PP2 and N = 5 for all the other conditions. Data represented as mean ± s.e.m. *P* values calculated by ordinary one-way ANOVA with Sidak's multiple comparisons test. (**C**) Colocalization (arrowheads) of EphB4 (green) and phospho-CAV1 (pCAV1, red) in HUVECs 30 min after stimulation with human IgG/Fc or ephrin-B2-Fc. Rightmost panels show higher magnification of selected areas. Cell junctions, VE-Cadherin (CDH5, white); nuclei, DAPI (blue). ns, not significant.

The online version of this article includes the following source data for figure 6:

**Source data 1.** Source data for *Figure 6B*.

the genes *Irs1*, *Rragd*, *Hif1a* and *Vegfa* were downregulated in mutant hearts (*Figure 5E,F*; *Figure 5—figure supplement 1B,C* and *Figure 5—source data 1*) (*Ronnebaum and Patterson, 2010*; *Schips et al., 2011*; *Sciarretta et al., 2018*; *Zhang et al., 2010*).

To gain a more complete overview of the alterations in Ephb4^ΔEC hearts, we analyzed the changes at the protein level by mass spectrometry of ventricles relative to littermate controls (*Figure 5—figure supplement 1D–G*). This revealed a total of 455 significant altered proteins (p<0.05), of which 206 were upregulated and 249 downregulated (*Figure 5—figure supplement 1E* and

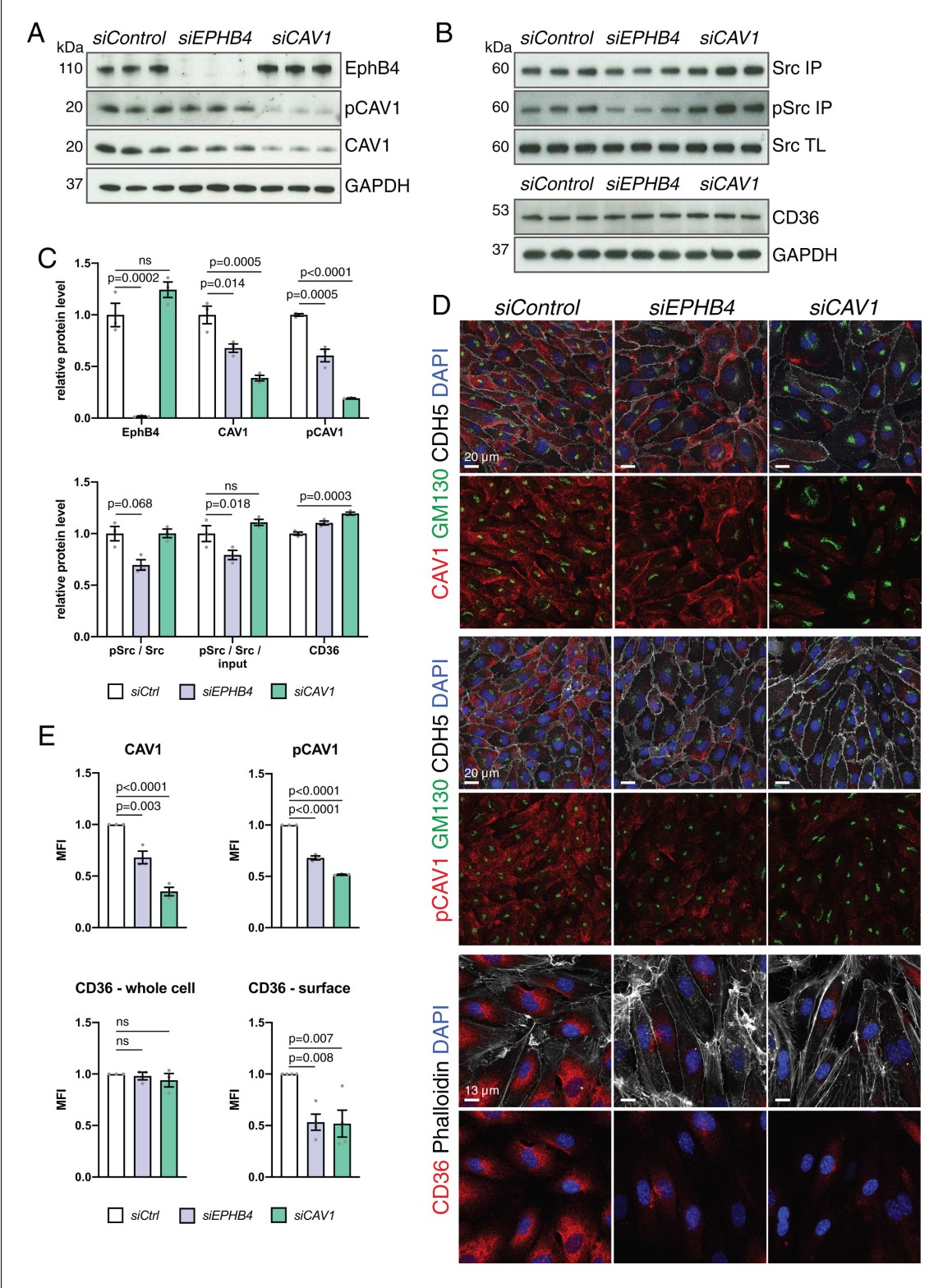

**Figure 7.** Alterations after knockdown of *EPHB4* expression in ECs. (**A**) Western blot analysis of HUVECs transfected with *siControl*, *siEPHB4* or *siCAV1*, as indicated. Knockdown cells showed a reduction of CAV1 and pCAV1 protein levels. N = 3 for all treatments. (**B**) Src input total cell lysate (TL) and tyrosine phosphorylation of immunoprecipitated (IP) Src in siRNA-treated HUVECs. Bottom panels show levels of CD36 and GAPDH (loading control). N = 3 for all treatments. (**C**) Quantitation of immunoblots for levels of EphB4, CAV1, pCAV1, CD36, pSrc/Src and ratio of pSrc/Src/ input. N = 3 for all treatments. Data represented as mean ± s.e.m. *P* values calculated by ordinary one-way ANOVA with Sidak's multiple comparisons test. (**D**) Confocal

*Figure 7 continued on next page*

*Figure 7 continued*

images of siRNA-transfected HUVECs stained with CAV1 or pCAV1 (red), GM130 (green), VE-Cadherin (CDH5; white) and nuclei (DAPI; blue). Bottom panels show surface CD36 (red), Phalloidin (white), and nuclei (DAPI; blue). (E) Quantitation of CAV1, pCAV1 and CD36 MFI per whole cell and of surface CD36 signal of immunostained HUVECs, as shown in (D). N = 3 experiments for CAV1 and pCAV1, in each of them 30 cells were quantified from three images (10 cells/ image). N=3 for CD36 surface staining, in which three images/experiment were quantified for all siRNA conditions. Data represented as mean ± s.e.m. *P* values calculated by ordinary one-way ANOVA with Sidak's multiple comparisons test. MFI, mean fluorescence intensity; ns, not significant.

The online version of this article includes the following source data and figure supplement(s) for figure 7:

**Source data 1.** Source data for *Figure 7C*.
**Source data 2.** Source data for *Figure 7E*.
**Figure supplement 1.** CD36 localization in *EPHB4* and *CAV1* knockdown HUVECs.
**Figure supplement 1—source data 1.** Source data for *Figure 7—figure supplement 1B*.

*Figure 5—source data 2*). Remarkably, GO analysis of biological processes in the mutant ventricles revealed an upregulation of metabolic processes (*Figure 5—figure supplement 1F*) accompanied by downregulation of endocytic recycling (*Figure 5—figure supplement 1G*). Altogether these results suggest that cardiac remodeling in Ephb4$^{\Delta EC}$ mutants is linked to defective transport processes and a metabolic shift towards glycolysis in heart.

## EphB4 controls caveolar function and fatty acid uptake

Pathways implicated in signal transduction downstream of EphB4, such as signaling through Phosphoinositide 3-kinase (PI3K)-Akt (*Steinle et al., 2002*), mitogen-activated protein kinase (MAPK)/ERK (*Das et al., 2010*) and Src (*Yang et al., 2006*), are also known to regulate intercellular vesicular trafficking (*Bhattacharya et al., 2016*). In cultured human umbilical vein endothelial cells (HUVECs), stimulation with recombinant and soluble ephrin-B2/Fc protein induces phosphorylation of Akt and Src, which is blocked by the pathway-specific inhibitors LY294002 and PP2, respectively (*Figure 6A, B*). In contrast, levels of phosphorylated ERK1/2 were reduced after ephrin-B2/Fc treatment. Stimulation with EphB4/Fc, which is known to activate signaling through the cytoplasmic domain of ephrin-B2 (*Palmer et al., 2002*), led to activation of Src but not of Akt and ERK1/2 in cultured HUVECs (*Figure 6A,B*). We also noted overlap between EphB4 receptor, detected by binding of ephrin-B2/Fc, and phosphorylated caveolin 1 (CAV1) (*Figure 6C*), an integral membrane protein in caveolar membranes and regulator of caveolae-mediated endocytosis. Interestingly, previous work has linked phosphorylation of caveolin to Src kinase (*Li et al., 1996*; *Sverdlov et al., 2007*).

Next, we treated HUVECs with an siRNA against *EPHB4 (siEPHB4)* or an unrelated negative control. Transfection of *siEPHB4* efficiently abolished expression of the receptor tyrosine kinase at protein level, as assessed by Western blotting (*Figure 7A,C*). Knockdown of *EPHB4* also resulted in significant reductions of phospho-Src and phosphorylated CAV1, the latter of which was also profoundly reduced in HUVECs treated with *siCAV1* siRNA (*Figure 7A–C*). CAV1 and phospho-CAV1 immunostaining decorate the cytoplasm and the Golgi apparatus, identified by anti-GM130 antibody signal, and both CAV1 and phospho-CAV1 were significantly reduced after knockdown of *EPHB4* or *CAV1* (*Figure 7D,E*). *siEPHB4* or *siCAV1* transfection of HUVECs also led to a significant increase in cell size (*Figure 8—figure supplement 1A*).

Further arguing for EphB4-induced phosphorylation of CAV1 through Src, ephrin-B2/Fc treatment of HUVECs significantly increased phospho-CAV1, which was blocked by addition of the Src family kinase inhibitor PP2 (*Figure 8A,B*). In contrast, EphB4/Fc treatment did not lead to appreciable changes in phospho-CAV1 (*Figure 8A,B*). In immunostainings, stimulation with ephrin-B2/Fc increased phospho-CAV1 signals, which was blocked by presence of PP2 (*Figure 8C–E*). In contrast, inhibition of PI3K-Akt with LY294002 or of MAPK/ERK with U0126 had no significant effect on CAV1 phosphorylation (*Figure 8D,E* and *Figure 8—figure supplement 1C*). Treatment with ephrin-B2/Fc or EphB4/Fc alone or in combination with PP2, LY294002 or U0126 did not lead to alterations in cell size (*Figure 8—figure supplement 1B*). Confirming that the effect of ephrin-B2/Fc is indeed mediated by EphB4 and not another member of the large Eph receptor family, the increase in CAV1 phosphorylation after ephrin-B2 stimulation was suppressed in HUVECs treated with *siEPHB4* (*Figure 8G*). Together, these results argue that EphB4 activation triggers the phosphorylation of CAV1 through the cytoplasmic kinase Src, which generates a molecular link to caveolar transport in

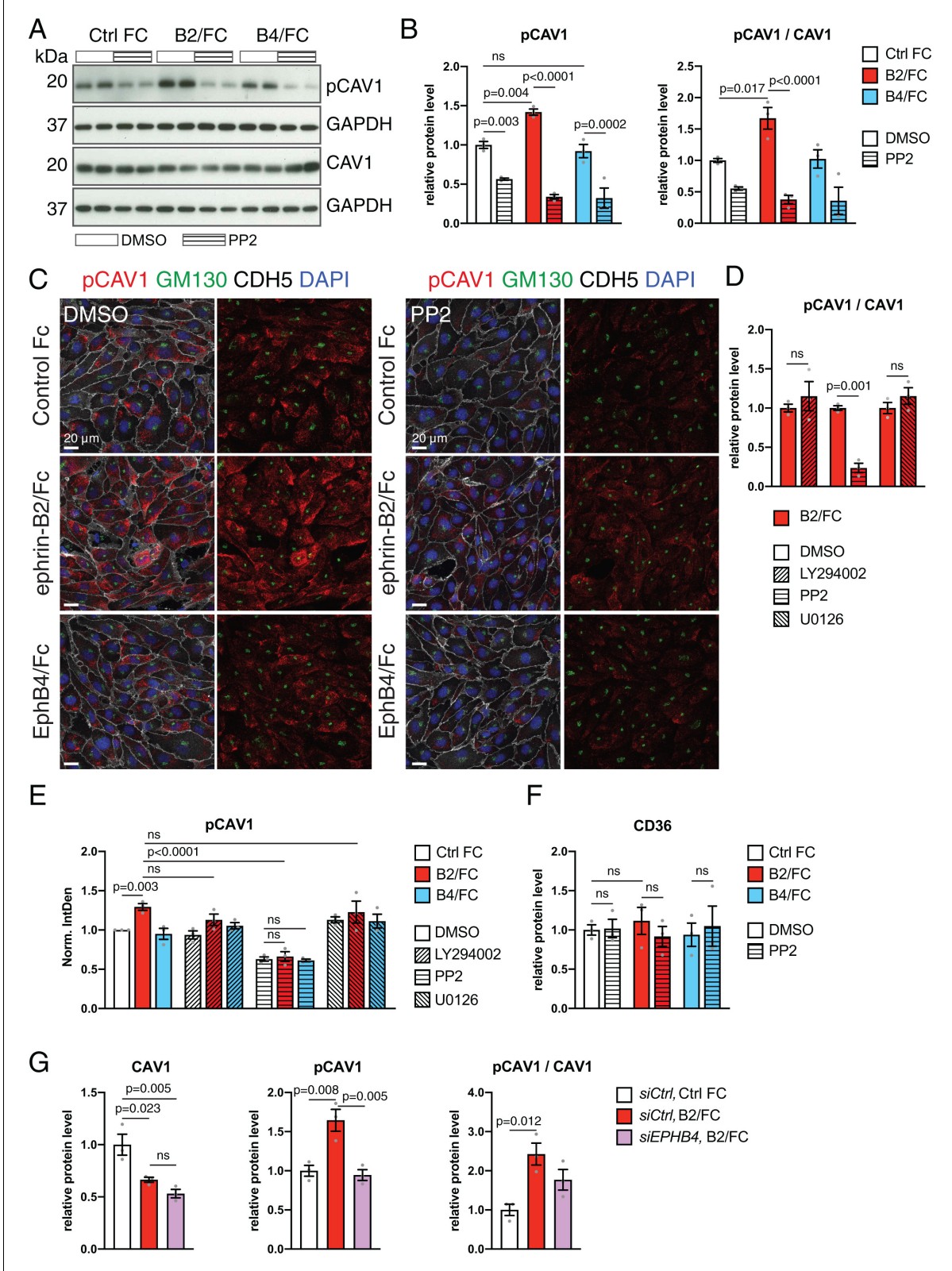

**Figure 8.** EphB4 activation induces Src-dependent phosphorylation via Src. (**A**) Western blot analysis of CAV1 and pCAV1 levels in HUVECs stimulated with control human IgG/Fc (Ctrl Fc), ephrin-B2/Fc (B2/Fc) or EphB4/Fc (B4/Fc) (4 µg/ml, preclustered with 10µg/ml goat anti-human IgG) in combination with DMSO (vehicle control) or the Src inhibitor PP2. GAPDH is shown as loading control. (**B, F**) Quantitation of Western blots results (see **A**) for pCAV1 together with the ratio pCAV1/CAV1 (**B**) and levels of CD36 (**F**). N = 3 for all treatments. Data represented as mean ± s.e.m. *P* values calculated by

*Figure 8 continued*

ordinary one-way ANOVA with Sidak's multiple comparisons test. (**C**) Confocal images of HUVECs after stimulation with Fc proteins in combination with DMSO or PP2. Stainings show pCAV1 (red), GM130 (green), CDH5 (white) and DAPI (blue). (**D**) Ratio of pCAV1/CAV1 in lysates from HUVECs treated with ephrin-B2/Fc in combination with DMSO (vehicle control) or the indicated inhibitors (LY294002, PP2 and U0126). N = 3 for all treatments. Data represented as mean ± s.e.m. *P* values calculated by unpaired two-tailed *t* test with Welch's correction. (**E**) Quantitation of pCAV1 immunosignal per cell (as shown in C) of HUVECs stimulated with Fc proteins along with inhibitor treatment. N = 3 experiments, in each of them 30 cells were quantified from three images (10 cells/ image) for all treatments. Data represented as mean ± s.e.m. *P* values calculated by ordinary one-way ANOVA test. (**G**) Quantitation of normalized CAV1, pCAV1 and pCAV1/CAV1 in immunoblotted lysates from HUVECs treated with control *siCtrl* or *siEPHB4* in combination with Ctrl Fc or ephrin-B2/Fc (B2/Fc). N = 3 for all conditions. Data represented as mean ± s.e.m. *P* values calculated by ordinary one-way ANOVA. Norm. IntDen, normalized integrated density; ns, not significant.

The online version of this article includes the following source data and figure supplement(s) for figure 8:

**Source data 1.** Source data for *Figure 8B,F*.
**Source data 2.** Source data for *Figure 8D*.
**Source data 3.** Source data for *Figure 8E*.
**Source data 4.** Source data for *Figure 8G*.
**Figure supplement 1.** Effect of Akt, ERK and Src inhibition on CAV1 and CD36 localization in stimulated HUVECs.
**Figure supplement 1—source data 1.** Source data for *Figure 8—figure supplement 1A,B*.
**Figure supplement 1—source data 2.** Source data for *Figure 8—figure supplement 1C*.

ECs and provides an explanation for the observed accumulation of caveolae at the basolateral membrane of the Ephb4$^{\Delta EC}$ cardiac endothelium.

Next, we analyzed caveolae and clathrin-mediated endocytosis by assessing the uptake of fluorescent-labeled albumin and transferrin, respectively (*Hopkins and Trowbridge, 1983*; *Iacopetta et al., 1983*; *Li et al., 2013*). Flow cytometry and the analysis of stained cells revealed significant reductions in the uptake of albumin, a caveolae-dependent process, upon EphB4 downregulation (*Figure 9A,C* and *Figure 9—figure supplement 1A,B*). Similarly, transfection with *siCAV1* impaired the uptake of albumin (*Figure 9A,C*). In contrast, the clathrin-controlled uptake of transferrin was unaffected in non-permeabilized *siEPHB4* cells and slightly reduced after permeabilization and immunostaining (*Figure 9—figure supplement 1C–F*).

To address whether impairment of caveolae might affect fatty acid transport in HUVECs, we exposed cultured cells to BODIPY C$_{12}$, a fluorescent fatty acid analog mimicking properties of natural lipids (*Bai and Pagano, 1997*). This approach revealed significantly reduced fatty acid uptake in *siEPHB4* HUVECs relative to control cells, based on both flow cytometry data and the analysis of stained cells (*Figure 9B,C* and *Figure 9—figure supplement 1G,H*). CD36 is a membrane receptor mediating the endocytosis of fatty acids and requires caveolar transport for its correct localization and function (*Ring et al., 2006*). While immunostaining of *siEPHB4* and *siCAV1* transfected HUVECs revealed no change in total CD36 immunosignal relative to control cells (*Figure 7E* and *Figure 7—figure supplement 1A*), surface CD36 was significantly reduced relative to control cells (*Figure 7D, E*). Further, CD36 localization to Golgi was increased after knockdown of *EPHB4* or *CAV1* (*Figure 7—figure supplement 1A,B*), potentially indicating CD36 synthesis and thereby a compensatory response to reduced fatty acid uptake. CD36 levels were not significantly altered after acute stimulation of HUVECs with ephrin-B2/Fc (*Figure 8F*). In vivo, CD36 immunosignal is downregulated in capillaries of the Ephb4$^{\Delta EC}$ myocardium but appears upregulated in the membrane of cardiomyocytes, again suggesting a potential compensatory mechanism to restore fatty acid uptake (*Figure 9D*).

Given the known link between metabolic changes and heart remodeling in human cardiomyopathies, we also assessed metabolic changes in Ephb4$^{\Delta EC}$ mice in vivo. Glucose concentration was reduced in mutant plasma relative to littermate controls, whereas levels of triglycerides and free fatty acids appeared unaffected (*Figure 9—figure supplement 2A*). Interestingly, histological analysis of Ephb4$^{\Delta EC}$ livers revealed the accumulation of fat (*Figure 9—figure supplement 2B*), suggesting that defective caveolar transport and fatty acid uptake by mutant cardiac ECs is accompanied by ectopic lipid deposition in liver.

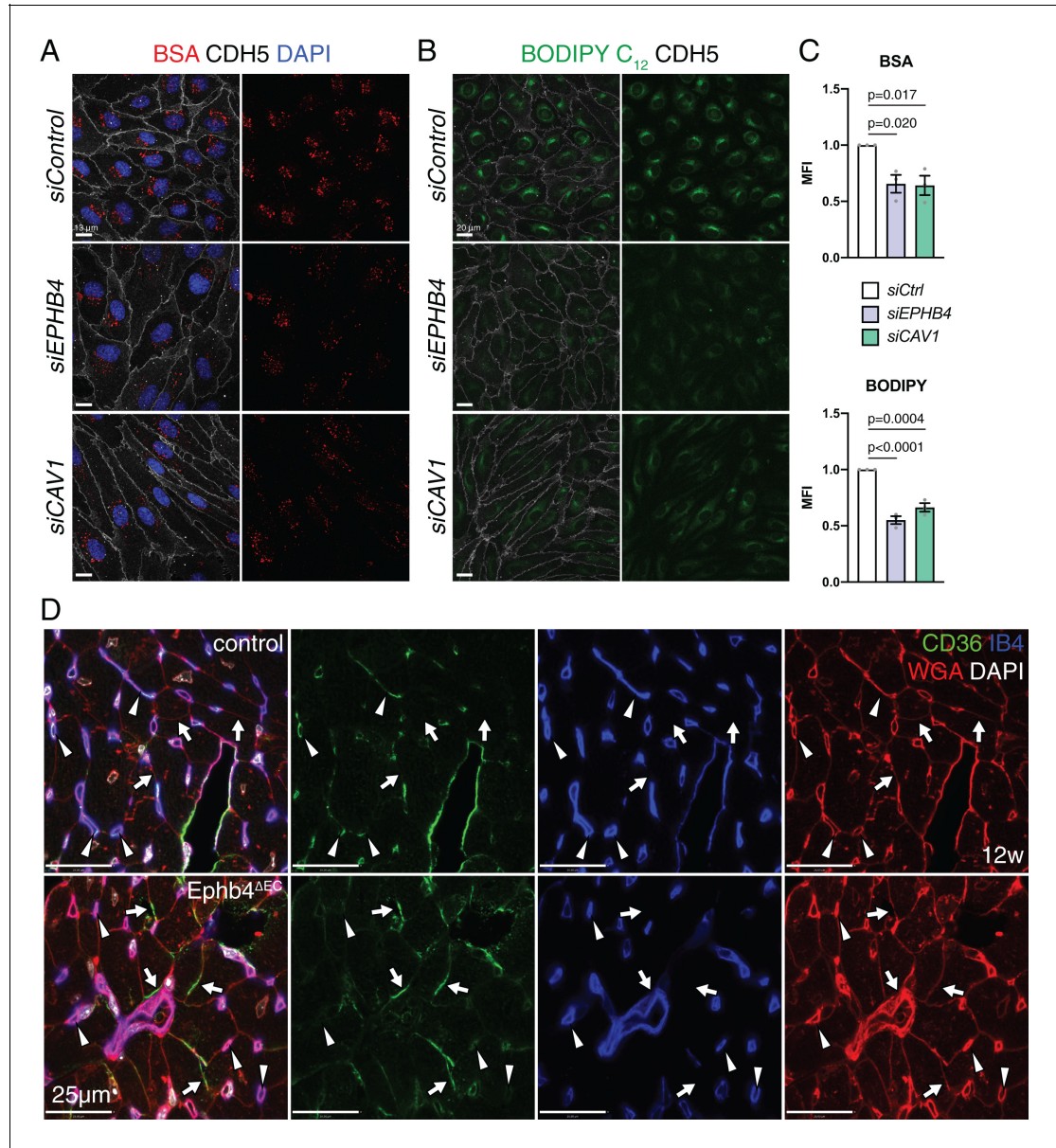

**Figure 9.** EphB4 is required for caveolar transport. (A) Confocal images of HUVECs transfected *siControl*, *siEPHB4* and *siCAV1*. Cells were fixed 30 min after exposure to BSA-555 (red) and immunostained with anti-CDH5 (white) antibody and DAPI (nuclei, blue). (B) Confocal images of *siControl*, *siEPHB4* and *siCAV1* HUVEC cells fixed after 30 min of treatment with BODIPY $C_{12}$-500/510 (green) and immunostained for CDH5 (white). (C) Quantitation of BSA-555 (top) and BODIPY $C_{12}$-500/510 (bottom) MFI per cell in *siControl*, *siEPHB4* and *siCAV1* HUVECs. Note reduced uptake of BSA and BODIPY in *siEPHB4* and *siCAV1* cells. N = 3 experiments, in each of them 30 cells were quantified from three images (10 cells/ image) for all knockdown conditions. Data represented as mean ± s.e.m. *P* values calculated by ordinary one-way ANOVA test. (D) CD36 expression in 12 week-old control and Ephb4$^{ΔEC}$ sectioned heart. Panels show details of the inner part of the left ventricle. CD36 immunosignal is reduced in mutant capillaries (arrowheads) but increased in the membrane of cardiomyocytes (arrows). MFI, mean fluorescence intensity.

The online version of this article includes the following source data and figure supplement(s) for figure 9:

**Source data 1.** Source data for *Figure 9C*.
**Figure supplement 1.** BSA, transferrin and BODIPY uptake in HUVECs.
**Figure supplement 1—source data 1.** Source data for *Figure 9—figure supplement 1A,C,F,G*.
**Figure supplement 2.** Metabolic features of Ephb4$^{ΔEC}$ mice.
**Figure supplement 2—source data 1.** Source data for *Figure 9—figure supplement 2A*.

## EphB4 maintains the structural integrity of the cardiac endothelium

Next, we investigated the cause of the ruptures and microhemorrhages in the Ephb4$^{\Delta EC}$ cardiac endothelium. Caveolar transport is also required for the recycling and proper function of focal adhesion and cell junction proteins in ECs (*Nethe and Hordijk, 2011*). Moreover, the junction protein VE-Cadherin together with linker proteins, such as Vinculin, couples cell junctions to the actin cytoskeleton (*Abu Taha and Schnittler, 2014*) and thereby maintains the integrity of the EC monolayer, which is probably most relevant in environments characterized by high mechanical loading such as the heart. In cultured HUVECs, *siEPHB4* transfection reduced the accumulation and localization of focal adhesion proteins in cell junctions under static conditions, whereas VE-Cadherin localization was maintained (*Figure 10A,C*). When HUVECs were cultured under cyclic tension mimicking 10% of sinusoidal stretch for 24 hr, control cells were able to maintain a continuous monolayer but *siEPHB4*

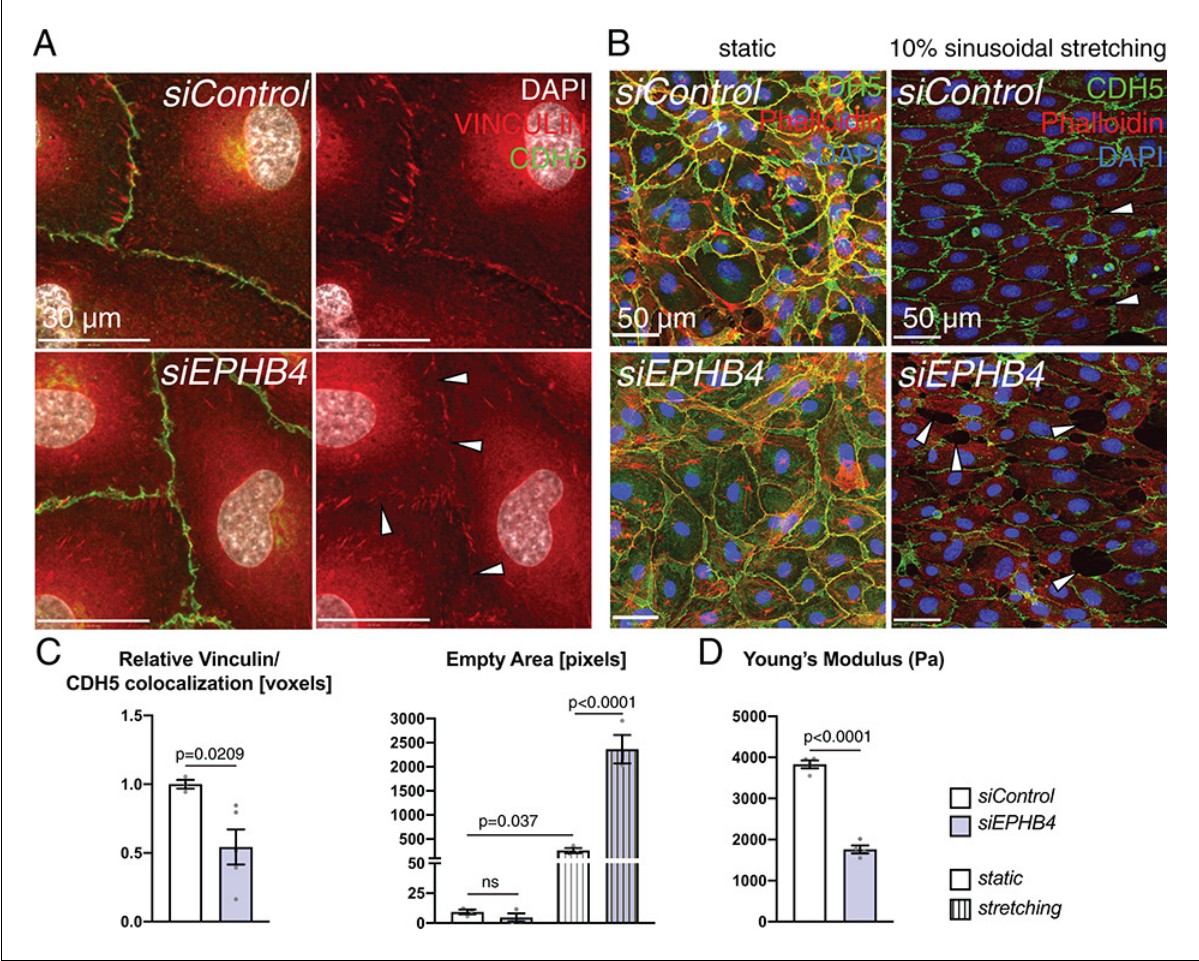

**Figure 10.** EphB4 is required for endothelial integrity. (**A**) Immunohistochemistry of cultured HUVECs. Relative colocalization of Vinculin and CDH5 is reduced in the cell contact area (arrowheads) of *siEPHB4* cells. (**B**) Mechanical stress assay on cultured HUVECs transfected with *siEPHB4*. Cells are able to form a monolayer on a membrane under static conditions. Sinusoidal stretch of the membrane disrupts the monolayer, as measured by the increase of empty area in *siEPHB4* HUVEC cells (arrowheads) (**C**). (**C**) Quantitation of relative colocalization of Vinculin and CDH5 (left). N = 3 for *siControl* and N = 5 for *siEPHB4*. Quantitation of the empty area between *siControl* and *siEPHB4* HUVECs in static conditions or with stretching (right). N = 3. Data represented as mean ± s.e.m. *P* value calculated by unpaired two-tailed *t* test with Welch's correction (left) or ordinary one-way ANOVA with Sidak's multiple comparisons test (right). (**D**) Atomic force microscopy analysis of cell stiffness. Young's modulus is reduced in *siEPHB4* cells. Each dish contains three force maps that equal 768 force-distance curves. Studied areas (150 × 150 µm) contain 10–16 cells. In between 16 and 25 force-distance curves were measured per cell. N = 4 areas. Data represented as mean ± s.e.m. *P* values calculated by unpaired two-tailed *t* test with Welch's correction. ns, not significant.

The online version of this article includes the following source data for figure 10:

**Source data 1.** Source data for *Figure 10C,D*.

cells were frequently separated by large gaps (*Figure 10B,C*). Furthermore, atomic force microscopy (AFM) revealed that the downregulation of *EPHB4* results in decreased cell stiffness relative to control HUVECs, as indicated by a significant reduction in the Young's Module that measures membrane displacement upon force application with a cantilever (*Figure 10D*). Altogether, these results show that the EphB4 receptor is required for the structural integrity of cultured ECs, which may well explain the vessel wall ruptures observed in vivo.

In conclusion, our results show the loss of EphB4 impairs the mechanical properties and transport function of ECs, which results in capillary ruptures, metabolic alterations, cardiomyocyte hypertrophic remodeling, and heart dilation.

## Discussion

There is increasing evidence that blood vessels acquire organ-specific specialization (*Augustin and Koh, 2017*), which is necessary to deal with specific metabolic needs and other, typically local properties of the surrounding tissue. In the case of the beating heart, the vasculature has to resist extensive mechanical strain and needs to provide cardiomyocytes with sufficient amounts of fatty acids. We propose that these special properties of the cardiac vasculature involve interactions between the receptor tyrosine kinase EphB4 and its ligand ephrin-B2. While ephrin-B2 and EphB4 are frequently used as markers for arteries and veins, respectively, both molecules are also co-expressed, albeit at lower levels, in cardiac capillaries. This is consistent with other reports emphasizing important roles of these molecules in sprouting vessels and capillaries (*Groppa et al., 2018*; *Sawamiphak et al., 2010*; *Wang et al., 2010*). In line with the close proximity of capillaries and cardiomyocytes, EC-specific inactivation of the *Ephb4* gene induces a hypertrophic response in the latter, which triggers heart remodeling and gives rise to a phenotype that resembles human dilated cardiomyopathy in its features and development. While EphB4 signaling is required for nutrient transport and the structural integrity of coronary capillaries, it is noteworthy that oxygen transport to the heart is not appreciably compromised despite vascular ruptures and microhemorrhaging. At the molecular level, defects in the *Ephb4*-deficient endothelium are linked to compromised caveolar function and reduced CAV1 phosphorylation, which involves the kinase Src, a known mediator of Eph receptor signaling (*Liang et al., 2019*; *Lisabeth et al., 2013*). This is also consistent with a previous report showing that stimulated EphB4 associates with CAV1 and induces its phosphorylation at Tyr-14 (*Muto et al., 2011*). Src-dependent phosphorylation of this residue in CAV1 promotes swelling of caveolae and their release from the plasma membrane (*Zimnicka et al., 2016*). CAV1 has also been shown to be required for signaling by EphB1, a receptor that is closely related to EphB4 (*Vihanto, 2006*).

Caveolae have been associated with cardiac disease previously. CAV1 has been reported to be cardioprotective (*Das et al., 2012*) and its absence is associated with cardiac hypertrophy (*Cohen et al., 2003*). *Cav3* knockout mice also show a cardiomyopathy characterized by hypertrophy, heart dilation and reduced contractibility (*Woodman et al., 2002*). *Cav1 Cav3* double knockout animals, which completely lack caveolins because CAV2 is degraded in the absence of CAV1, develop cardiac hypertrophy and contractile failure (*Park et al., 2002*). In the healthy heart, caveolar transport is required for the membrane translocation and correct function of fatty acid translocase FAT/CD36 (*Ring et al., 2006*; *Son et al., 2018*), which mediates the uptake of fatty acids. Fatty acids will be delivered to the heart muscle cells that use them to obtain about 50% to 70% of their energy (*Lopaschuk et al., 2010*). Although CD36 is also expressed in the cardiomyocytes and its absence causes reduced fatty acids uptake and utilization by muscle cells (*Coburn et al., 2000*), it is endothelial CD36 that is critical for the uptake of fatty acids by the heart (*Son et al., 2018*). Accordingly, we found that CD36 in the plasma membrane, a localization that is critical for the function of the fatty acid translocase (*Glatz and Luiken, 2018*), is profoundly reduced in cultured ECs after knockdown of *EPHB4* expression. CD36 is also reduced in Ephb4$^{\Delta EC}$ coronary capillaries in vivo, whereas upregulation of CD36 in mutant cardiomyocytes might be a compensatory effect due the lack of fatty acids provided by the endothelium. In line with the results described above, CD36-deficient mice have increased fatty acid uptake in the liver and develop fatty liver similar to Ephb4$^{\Delta EC}$ mutants (*Goudriaan et al., 2003*). Alterations in myocardial metabolic substrate have been related to heart failure and dilated cardiomyopathy (*Neubauer et al., 1997*) and it is noteworthy that DCM patients show CD36 deficiency (*Tanaka et al., 1997*).

Our findings also show that EphB4 signaling is required for maintaining the structural integrity of coronary capillaries. In this context, caveolae-mediated trafficking has been shown to be crucial for the turnover of focal adhesion points (*Nethe and Hordijk, 2011*). Proteins such as Vinculin or Paxillin form a molecular bridge, which couples the cytoskeleton to membrane proteins such as integrins (*Burridge and Chrzanowska-Wodnicka, 1996*; *Ezzell et al., 1997*) and thereby confers stiffness and mechanical stability to cells (*Matthews et al., 2004*). Mechanical stability is especially important in the heart, an organ where the vasculature is exposed to constant mechanical stress due to heartbeat (*Thorin and Thorin-Trescases, 2009*). Our results suggest that EphB4, presumably due to its role in caveolar trafficking, is required for the maintenance of focal adhesion points and thereby the structural integrity of ECs exposed to mechanical forces.

Taking all together, we propose that EphB4 and its ligand ephrin-B2 maintain fundamental properties of the endothelium in the adult heart. This function is distinct from the well-established roles of these molecules in the regulation of angiogenic blood vessel growth (*Adams et al., 1999*; *Gerety and Anderson, 2002*; *Gerety et al., 1999*; *Salvucci et al., 2006*; *Wang et al., 1998*). As the expression of EphB4 and ephrin-B2 is not confined to the cardiac vasculature, it remains unclear how these molecules can endow vessels with specialized, organ-specific properties. Previous work has shown that Eph/ephrin molecules and, in particular, ephrin-B2 are important regulators of intracellular trafficking processes such as the internalization of receptors from the cell surface (*Essmann et al., 2008*; *Gaitanos et al., 2016*; *Sawamiphak et al., 2010*; *Wang et al., 2010*). Thus, it is possible that Eph/ephrin interactions control fundamental processes, such as endocytosis and vesicular trafficking, in many different cellular settings, while the local biological outcome may strongly depend on the expression and function of specific cargo molecules. In addition, other regulators of caveolae, such as integrins (*Echarri et al., 2007*) might be more critical in other organs, which could contribute to local, tissue-specific properties of the vasculature.

Irrespective of the open questions outlined above, our work establishes that defects in the coronary endothelium can result in defects resembling idiopathic cardiomyopathy. Mutations in the human *EPHB4* gene are associated with various pathologies, such Capillary Malformation-Arteriovenous Malformation 2 (*Amyere et al., 2017*) and vein of Galen aneurysmal malformations (*Vivanti et al., 2018*), which are known to be associated with heart failure. Accordingly, vascular defects should be considered in the diagnosis and, potentially, also treatment of cardiac disease in the future.

## Materials and methods

The Key Resources Table (*Supplementary file 1*) provides a list with the mouse strains, cell line, antibodies, reagents, kits and software used for this study.

### Mouse strains

Mouse alleles used in this manuscript are listed including their bibliographic references. *Ephb4^flox* (*Wang et al., 2015*), *Efnb2^flox* (*Grunwald et al., 2004*), *Cdh5^CreERT2* (*Wang et al., 2010*), *Prox1-CreERT2* (*Bazigou et al., 2011*), *Rosa26R-mTmG* (*Muzumdar et al., 2007*), and *Efnb2::GFP* (*Davy and Soriano, 2007*). Mice were genotyped as previously described. All animal procedures were performed according to relevant laws and institutional guidelines, were approved by local animal ethics committees and were conducted with permissions (84–02.04.2015.A185, 84–02.04.2017.A238, 81–02.04.2019.A114) granted by the Landesamt für Natur, Umwelt und Verbraucherschutz (LANUV) of North Rhine-Westphalia.

### Loss-of-function genetics

*Cdh5^CreERT2* and *Prox1^CreERT2* mice were bred into a background of animals carrying *loxP*-flanked *Ephb4* and *Efnb2* gene. Experiments were performed only in male adults. Cre activity was induced by an injection of 100 µl 5 mg/ml 4-hydroxy tamoxifen (Sigma H7904) solution every day during 5 consecutive days starting at 8 weeks of age. Animals were then analyzed either at 10 weeks or 12 weeks of age.

## Cardiac Magnetic Resonance Imaging (CMRI)

CMRI was performed on a 9.4 Tesla BioSpec 94/20 system (Bruker Biospin, Ettlingen, Germany) using a 35 mm quadrature birdcage coil (Rapid Biomedical, Rimpar, Germany).

Mice were anesthetized in a chamber with 3% isoflurane in oxygen (flow rate 1 L/min). During the measurements, anesthesia was maintained at 1–2% isoflurane in a mixture of medical and oxygen (3:7 at 1 L/min) through a customized anesthesia mask. Rectal temperature and breathing rate were monitored throughout the measurement and body temperature was maintained at 36–37°C using a warm water filled heating pad.

For analyzing the cardiac function and myocardial mass, a stack of contiguous short-axis slices was acquired to cover the entire right and left ventricle. A retrospectively -gated cine FLASH (Intra-Gate FLASH, Bruker) sequence was used with the following parameters: number of slices: 9–10, slice thickness: 1 mm, matrix: $232 \times 232$, field of view: $30 \times 30 \text{ mm}^2$, repetition time: 5.0 ms, echo time: 2.52 ms, flip angle: 15°, number of repetitions: 200, scan time per slice: 1 min 58 s. IntraGate FLASH Data were reconstructed using Paravision software (Bruker) to obtain cinematic movies with 20 images per heart cycle and slice.

The end-diastolic and end-systolic heart phase of each slice was identified visually, using the ImageJ software (Rasband, W.S., ImageJ, U. S. National Institutes of Health, Bethesda, Maryland, USA, https://imagej.nih.gov/ij/, 1997–2016.). End-diastolic and end-systolic stacks were assembled, and analyzed using the Amira software (Visage Imaging GmbH, Berlin, Germany). Volume and wall thickness of the left and right ventricular for both time points were measured semi-automatically and the following functional cardiac parameters were calculated: wall thickness and left and right ventricular mass.

## Echocardiography

Mice were anaesthetized with isoflurane (1.5% in oxygen). The chest was shaved and 2D-guided B- and M-mode echocardiography was performed using a Vevo2100 (FUJIFILM VisualSonics, Inc, Toronto, Canada) equipped with an 22-55MHz linear array transducer to derive LV functional and dimensional parameters. Long- and short-axis views were obtained as described previously (*Stypmann et al., 2009*).

## Tissue processing

Heart, skeletal muscle and liver were dissected in cold PBS and fixed overnight in 4% PFA at 4°C. Samples were then processed for cryo-sectioning by cryo-protecting them with a series of consecutive overnight washes in PBS containing increasing concentration of sucrose (Sigma, S0389) (10%, 20% and 30%) and finally embedded in embedding solution (15% sucrose, 8% gelatin (Sigma, G1890) and 1% polyvinylpyrrolidone (pvp; Sigma, P5288) in PBS). After waiting for the embedding solution to solidify, samples were frozen at −80°C. Tibias were fixed overnight in 2% PFA at 4°C and previous to its embedding were decalcified with two consecutives overnight 15% EDTA washes at 4°C. Then cryoprotected with a single wash of 20% sucrose and 1% pvp in PBS, and finally embed as described above.

Samples were sectioned 50 μm thick in the cryostate Leica CM3050 S and collected into microscopy slides. Liver sections for Oil Red-O staining were cut 7 μm thick. After sectioning, samples were let dry overnight at room temperature and stored at −20°C.

## Cryo-section immunostaining

Before starting the immunostaining, the microscopy slide containing the samples was allowed to acquire room temperature. Slides were washed twice with PBS for 5 min to rehydrate the samples. Tissue was permeabilized with PBS containing 0.3% Triton X-100 by washing them three times 10 min and blocked for at least one hour in blocking solution (3% BSA, 0.1% Triton X-100% and 0.5% donkey serum in PBS). Primary antibodies were incubated overnight at 4°C in blocking solution. The consecutive day, four 5 min washes of PBS were performed to wash the primary antibodies. Secondary antibodies were incubated 1 hr at room temperature in PBS containing 5% of BSA. Immunostainings were imaged in a Leica SP8 confocal inverted microscope.

Primary antibodies: mouse anti-IB4-Biotin (Vector, B-1205, 1:25), WGA-tetramethylrhodamine (Invitrogen, W849, 1:100), chicken anti-GFP (2BScientific, GFP-1010, 1:100), rat anti-EphB4 (Hycult

Biotechnology, HM1099, 1:100), goat anti-EphB4 (R and D Systems, AF446, 1:100) rat anti-Icam2 (BD Pharmingen, 553326, 1:100), mouse anti-SMA-cy3 (Sigma, C6198, 1:100), rat anti-PdgfrB (eBioscience, 14-1402-82, 1:100), rat anti-Ter-119 (R and D, MAB1125, 1:100), rabbit anti-Vegfr3 (ReliaTech, 102-PA22S, 1:50), rabbit anti-Collagen Type I (Millipore, AB765P, 1:100), goat anti-Collagen Type IV (Millipore, AB769, 1:100) and rabbit anti-ERG (Abcam ab110639, 1:100).

Secondary antibodies: anti-chicken Alexa Fluor-488 (Jackson Immuno Research, 103-545-155, 1:500), anti-rabbit Alexa Fluor-488 (Invitrogen, A21206, 1:500), anti-rat Cy3 (Jackson Immuno Research, 712-165-150, 1:100), anti-rabbit Alexa Fluor-546 (Invitrogen, A10040, 1:500), anti-goat Alexa Fluor-488 (Invitrogen, A11055, 1:500), anti-goat Alexa Fluor-568 (Invitrogen, A11057, 1:500) and anti-rat coupled with horseradish peroxidase (Jackson Immuno Research, 712-035-153, 1:100). The signal was then amplified using a tyramide amplification kit coupled to Cy3 (Perkin Elmer, NEL744). IB4-Biotin signal was amplified using Streptavidin Alexa Fluor-488 (Invitrogen, S11223, 1:100) and Streptavidin Alexa Fluor-647 (Invitrogen, S32357, 1:100) depending on the other secondary antibody combination. Nuclei were counterstained using DAPI (Sigma, D9542).

Stainings were quantified using Volocity software version 6.3 (Perkin Elmer).

## Vibratome section immunostaining

Six littermate mice (3 controls and 3 mutants), 12 week-old, were anesthetized with ketamine (Zoetis, 344771)/Rompum (Bayer Healtcare, D-51368), and an incision was made along the chest cavity to expose the heart. Post perfusion with PBS and 4%PFA (PFA in PBS, 4°C, P6148, Sigma), heart, both the right and left kidneys, left lateral and right median liver were dissected out and placed in 2% PFA for 30 min. A vibrating blade microtome (VT1200, Leica) was used to section 150 μm thick sections of freshly dissected and fixed kidney and liver lobes. Sections were then refixed in 4% PFA at 4°C for 1 hr. After fixation and wash with PBS, sections were incubated in blocking buffer (0.1% BSA, 0.5% Triton-X 100, 5% donkey serum in PBS) at 4°C overnight. After five washes (0.1% Triton X-100/PBS) every 20 min, tissue slices were incubated with primary antibodies (diluted in blocking buffer) overnight at 4°C. Primary antibodies used were goat anti-CD31 (R and D Systems, AF3628, 1:100) and rat anti-ICAM2 (BD Pharmingen, 553326, 1:100). After washing to remove primary antibodies, a final overnight incubation (also at 4°C) with secondary antibodies and DAPI (diluted in blocking buffer) was carried out. Secondary antibodies used were donkey anti-goat Alexa Fluor-488 (Invitrogen, A11055, 1:500) and donkey anti-rat Alexa Fluor-647 (Jackson Immuno Research, 712-605-153, 1:500). Lastly, tissue sections were washed and then mounted using Fluoromount-G (Southern Biotech) on glass slides (with Secure-Seal spacers), placed under cover slips and sealed with nail polish at the edges.

Zeiss LSM880 confocal laser scanning microscope, in online fingerprinting mode, operating under ZEN software, was used to acquire immunofluorescence images of kidney and liver sections. Fiji ImageJ software was used for processing the images thus acquired and Illustrator (version CC 2018, Adobe) was used for compiling all the images. Within each experiment, laser excitation and confocal scanner detection were kept the same and all images were processed under identical parameters.

## Western blot

Heart ventricles were dissected and snap frozen in liquid nitrogen. Ventricles were homogenized in Lysis buffer (20 mM Tris-HCl pH 7.4, 1 mM EDTA, cOmplete ULTRA (Roche, 05892970001), 1% NP-40, 0.1% SDS, 150 mM NaCl) with Pestle (ARgos) and clarified by centrifugation at 20000 x g for 20 min at 4°C. Protein concentration in the lysates was measured using Precision Red Advanced protein assay reagent (Cytoskeleton). Soluble supernatants were prepared in SDS-PAGE sample buffer and analyzed by SDS-polyacrylamide gel electrophoresis and immunoblotting after loading 20 μg of total ventricle lysate. Signal was detected using horseradish peroxidase-conjugated secondary antibodies followed by ECL Prime or ECL detection reagent (GE Healthcare).

Cultured HUVEC cells were washed twice with PBS containing 1 mM PMSF and further incubated with RIPA modified buffer (20 mM Tris HCl, pH 8.0, 150 mM NaCl, 0.5% Triton X-100, 0.1% SDS, 0.1% Na-DOC, 2 mM EDTA, cOmplete ULTRA (Roche, 05892970001) and phosphatase inhibitor cocktail set V (EMD Millipore, 524629)) for 20 min at 4°C. Cells were scraped off and lysed by sonication. Lysates were spun down at 4°C for 10 min at full speed and protein concentration was quantified using BCA Protein Assay Kit (Pierce, 23225). Samples were prepared in SDS-PAGE sample

buffer and analyzed by SDS-polyacrylamide gel electrophoresis and immunoblotting after loading 1.5 µg of total cell lysate.

Primary antibodies: rabbit anti-GAPDH (Cell Signaling, 2118, 1:1000), goat anti-EphB4 (R and D Systems, AF446, 1:2000), rabbit anti-phospho-Akt (Ser473) (Cell Signaling, 4060, 1:1000), rabbit anti-Akt (Cell Signaling, 4691, 1:1000), rabbit anti-p44/42 MAPK (Cell Signaling, 4695, 1:1000), rabbit anti-phospho-p44/42 MAPK (The202/Thr204) (Cell Signaling, 4370, 1:1000), mouse anti-phospho-tyrosine (clone 4G10, Merck, 05–321, 1:1000), anti-rabbit Cav-1 (Cell Signaling, 3238, 1:50000), rabbit anti-phospho-Cav1 (Cell Signaling, 3251, 1:500), rabbit anti-Src (Cell Signaling, 2123, 1:1000), rabbit anti-CD36 (Abcam, ab133625, 1:1000), mouse anti-Vinculin (Sigma, V9131, 1:200) and mouse anti-beta-Actin (Santa Cruz, sc-47778, 1:6000).

Secondary antibodies: goat anti-rabbit IgG, HRP-linked whole Ab (Cell Signaling, 7074, 1:15000), sheep anti-mouse IgG, HRP-linked whole Ab (HG-Healthcare, NA931, 1:40000) and Peroxidase AffiniPure Bovine anti Goat IgG (H+L) (Jackson Immuno Research, 805-035-180, 1:15000).

Blots were quantified using ImageJ version 2.0.0 (Fiji) (*Schindelin et al., 2012*).

## Electron microscopy

Hearts from freshly killed 12 week-old mice were removed, cut into halves and fixed in 2% paraformaldehyde, 2% glutaraldehyde in 0.1M cacodylate buffer, pH 7.2. The left ventricle was cut in smaller cubes and subsequently post-fixed in 1% osmiumtetroxide, 1.5% potassiumferrocyanide in 0.1M cacodylate buffer, pH 7.2. The samples were stepwise dehydrated in ethanol, including en bloc 0.5% uranyl acetate staining during 70% ethanol incubation. Blocks were embedded in epon then sectioned ultrathin at 70 nm. Sections were collected on copper grids and stained with lead. The samples were analysed on a FEI-Tecnai 12 electron microscope (FEI, Eindhoven, The Netherlands) and representative areas were imaged with a 2 K CCD camera (Veleta, EMSIS, Münster, Germany).

## Hypoxia and cell death assay

Hypoxia was analyzed using Hypoxiprobe Plus kit-FITC (Hypoxiprobe, HP2). Animals were injected with 60 mg/kg of pimonidazole following recommended protocol. Anti-FITC-HRP signal was amplified using tyramide amplification kit coupled to Cy3 (Perkin Elmer, NEL744).

Cell death was analyzed using in situ Cell Death Detection Kit TMR red (Roche, 12156792910) according to manufacturer's instructions.

## RNA-seq

RNA was isolated from the ventricles of 12 week-old male mice. Sequencing libraries were prepared using TrueSeq Stranded RNA LT Kit Ribo-Zero Gold (Illumina, 15032619). The resulting mRNA libraries were sequenced on a MiSeq sequencer using 2 × 75 bp paired-end MiSeq v3 chemistry (Illumina). Sequenced reads were aligned to the *Mus musculus* reference genome GRCm38 (https://support.illumina.com/sequencing/sequencing_software/igenome.html) using TopHat2 (*Kim et al., 2013*). Then, aligned reads were used to quantify mRNA with HTSeq-count (*Anders et al., 2015*). Differential gene-expression analysis between control and *Ephb4* mutant mice was performed using DESeq2 (*Love et al., 2014*). Genes were considered as differentially expressed when the FDR-adjusted $P$ value was $\leq$ 0.05. Analysis were done using R (version 3.4.3; *R Development Core Team, 2013*; http://www.R-project.org/).

All RNA–seq data have been deposited in the ArrayExpress database at EMBL-EBI (www.ebi.ac.uk/arrayexpress) under the accession number E-MTAB-7686.

## Mass Spectrometry (MS)

Sample preparation and LC-MS/MS analysis: Shock frozen whole mouse ventricles (n = 3 for both genotypes) were homogenized at 4°C in 1 ml of lysis buffer, containing 6M guanidinium hydrochloride (GuHCL), 5 mM tris(2-carboxyethyl) phosphine (TCEP) and 10 mM chloroacetamide (CAA) using a Cryolys Evolution tissue homogenizer (Bertin Instruments; 2 ml CKM lysing kit, 3 × 23 s at 6500 rpm; 4°C). After centrifugation (10 min, 16000 g, 4°C), protein concentrations were determined by a Bradford assay (ThermoFisher Scentific, 23236) and 2 mg of total protein per heart were subjected first to a lysyl endopeptidase predigest (enzyme: protein ratio 1:100; 3 hr, 37°C), followed by dilution

to 2M GuHCl and trypsin digest overnight at 37°C (enzyme: protein ratio 1:100). Samples were then desalted using reversed-phase C18 Sep-Pak classic cartridges (Water) and lyophilized.

Peptides were dissolved in 1 ml Buffer A (10 mM NH$_4$OH, pH10.2) and subjected to offline high-pH reversed-phase prefractionation using a YMC-Triart C18 column (250 × 4.6 mm) on a Platin Blue high-pressure liquid chromatography system (Knauer) using a gradient from 0–5% B (90% acetonitril, 10 mM NH$_4$OH) in 1 min, from 5–39% B in 66 min, and from 39–67% in 5 min followed by a washout at 78% B and reequilibration at starting conditions. The instrument was operated at a flow rate of 1 ml/min. 47 fractions were collected, lyophilized and subjected to LC-MSMS analysis as described by *Bekker-Jensen et al. (2017)*. All samples were analysed on an Easy nLC 1200 system coupled to a Q Exactive HF mass spectrometer via a nanolectrospray source (ThermoFisher Scientific). Peptides were dissolved in buffer A (0.1% formic acid) and separated on a 25 cm column, in-house packed with 1.9 µm C18 beads (Reprosil -Pur C18 AQ, Dr. Maisch, Ammerbuch, Germany) using a gradient from 10–30% buffer B (80% acetonitril; 0.1% formic acid) within 25 min and from 30–45% in 5 min followed by a washout for 7 min at 90% B and re-equilibration at starting conditions (100% buffer A; flow rate 350 nl/min).

The Q-Exactive HF mass spectrometer was operated in data-dependent acquisition mode (spray voltage 2.1 kV; column temperature maintained at 45°C using a PRSO-V1 column oven (Sonation, Biberach, Germany)). MS1 scan resolution was set to 60,000 at m/z 200 and the mass range to m/z 350–1400. AGC target value was 3E6 with a maximum fill time of 100 ms. Fragmentation of peptides was achieved by Higher-energy collisional dissociation (HCD) using a top20 method (MS2 scan resolution 15.000 at 200 m/z; AGC Target value 1E5; maximum fill time 15 ms; isolation width 1.3 m/z; normalized collision energy 28). Dynamic exclusion of previously identified peptides was allowed and set to 30 s, singly charged and peptides assigned with charge of 6 and more were excluded from the analysis. Data were recorded with Xcalibur software (Thermo Scientific).

MS Data Analysis and label free quantification: Raw MS files were processed using the MaxQuant computational platform (version 1.6.2.6; *Cox and Mann, 2008*). Identification of peptides and proteins was enabled by the built-in Andromeda search engine by querying the concatenated forward and reverse mouse Uniprot database (UP000000589_10090.fasta; version from 12/2015) including common lab contaminants. The allowed initial mass deviation was set to 7ppm and 20ppm in the search for precursor and fragment ions, respectively. Trypsin with full enzyme specificity and only peptides with a minimum length of 7 amino acids was selected. A maximum of two missed cleavages was allowed; the 'match between runs' option was turned on. Carbamidomethylation (Cys) was set as fixed modification, while Oxidation (Met) and N - acetylation were defined as variable modifications. For peptide and protein identifications a minimum false discovery rate (FDR) of 1% was required. All mass spectrometry proteomics data have been deposited to the ProteomeXchange Consortium (http://proteomecentral.proteomexchange.org) (*Vizcaíno et al., 2014*) via the PRIDE partner repository with the dataset identifier PXD012575.

Relative label free quantification was based on the measurements of 3 independent biological replicates for control and Ephb4$^{\Delta EC}$ mice. Data processing was performed using the Perseus (version 1.6.2.1) (*Tyanova et al., 2016*). First, we eliminated from the MaxQuant output files the reverse and contaminant hits as well as hits that were identified by a modified site only. Proteins included in the analysis had to be identified with at least two peptides, one of which had to be unique for the protein group. Intensity values were logarithmized and missing values (NaN) were replaced by imputation, simulating signals of low abundant proteins within the distribution of measured values. A width of 0.3 SD and a downshift of 1.8 SD were used for this purpose. To identify in a supervised manner the sets of proteins that significantly distinguish the control group and the Ephb4$^{\Delta EC}$ proteomes, two-sample *t*-tests were performed using a p-value of 0.05. Principal component analysis was performed to project the proteome measurements into a two-dimensional data space. For this purpose, PCA was applied to all proteins that remained in the data set after filtering, transformation and imputation (6271 protein groups).

For the analysis, the primary gene identifier for each protein group was defined as the ones that represented the proteins which provided explanations for the all the peptides within a protein group.

## Gene Ontology

Gene ontology analysis was performed using the Enrichr platform (http://amp.pharm.mssm.edu/Enrichr/) (*Chen et al., 2013*; *Kuleshov et al., 2016*).

## Plasma analysis

Extracted blood was let to coagulate for 20 min at room temperature and then centrifugated 15 min at 3000 rpm at 4°C to separate blood plasma. Samples were pooled in groups of two replicates and sent for analysis to the Institut für Veterinarmedizinische Diagnostik, GmbH in Berlin.

## Liver oil Red-O

Samples were gently fixed with 4% PFA for 10 min. PFA was washed with running water for 10 min and 60% isopropanol for 2 min. Samples were incubated with Oil Red-O working solution for 15 min. The working solution was made by mixing 30 ml of stock solution (0.5 g Oil Red-O (Sigma, O0625) in 100 ml isopropanol) with 20 ml water, allowed to stand 10 min and then filtered. Working solution was washed for 2 min with 60% isopropanol and rehydrated with water three times for 5 min. Sections were counterstained with hematoxilin (Sigma, MHS16) for 4 min and then washed with water until it came clear. Finally, samples were mounted with aqueous mounting medium.

## Masson gold trichrome

Samples were fixed in 4% PFA and processed for cryo-sectioning. After sectioning samples were fixed again on Bouin's solution at room temperature overnight. Staining was performed using Masson Gold Trichrome kit (Roth, 3459.1) according to manufacturer's instructions.

## HUVEC cell culture, siRNA transfection, Fc stimulation and inhibitor treatment

HUVECs (ThermoFisher, C0035C) were cultured with EBM-2 endothelial cells medium (Lonza, CC-3156) supplemented with EGM-2 Single Quots (Lonza, CC-4176). Cells have not been tested for mycoplasma contamination. Cells were seeded in six well plates coated with 0.1% gelatin for protein extraction or in μ-Slide eight well (Ibidi, 80826) for staining and microscopy. Cells were transfected using Lipofectamine RNAiMAX procedure from Life technologies (Invitrogen, 13778). siRNAs: Negative control (Ambion, 4390844), *EPHB4* s244 (Ambion, 4390824) and *CAV1* (ThermoFisher, HSS141466). Four hours after transfection the medium was changed for fresh culture medium and left for 48 hr before analysis.

For Fc stimulation, cells were serum starved for 12 hr and then incubated in serum free medium containing 4 μg/ml of pre-clustered control human IgG (R and D Systems, 110-HG), human ephrin-B2/Fc (Biotechne, 7397-EB) or mouse EphB4/Fc (R and D Systems, 466-B4) proteins for 30 min. Proteins were preclustered immediately before treatment by incubating them in medium 30 min at 37°C with 0.2 μg anti-human IgG (Fc specific) (Jackson Immuno Research, 109-005-098) per μg of Fc protein at a final concentration of 10 μg/ml.

Inhibitor treatment was performed for 30 min, simultaneously with Fc stimulation or BODIPY uptake. Inhibitors used were dissolved in DMSO: LY294002 (final concentration 25 μM, Tocris, 1130), PP2 (final concentration 25 μM, Tocris, 1407) and U0126 (final concentration 10 μM, Promega, V1121). DMSO was used as vehicle control.

## Immunoprecipitation

To analyze pSrc level in HUVECs, Src protein immunoprecipitation was performed. Cultured HUVECs were either treated with siRNA or serum starved for 12 hr and further stimulated with preclustered Fc proteins (see above) for 30 min. Cells were washed twice with PBS containing 1 mM PMSF before lysis, which was performed with lysis buffer (50 mM Tris-HCl, pH 8.0, 100 mM NaCl, 2 mM EDTA, 0.2% IGEPAL CA-630, 0.5% Triton X-100, cOmplete ULTRA (Roche, 05892970001) and phosphatase inhibitor cocktail set V (EMD Millipore, 524629)). Lysis was further improved with an ultrasonic homogenizer (Sonicator UP100H). Lysates were spun down and protein concentration from the supernatants was measured using BCA Protein Assay Kit (Pierce, 23225). Equal protein amount in each lysate was further used for the immunoprecipitation. Input samples were further prepared in 2x SDS-PAGE sample buffer. Lysates were further precleared with 20 μl prewashed Protein G

Sepharose 4Fast Flow resin (GE Healthcare, 17-0618-01) for 20 min at 4°C. Supernatants were incubated with rabbit anti-Src antibody (Cell Signaling, 2123, 1:50), overnight at 4°C on the rotating wheel (15 rpm). As negative control normal rabbit IgG antibody (Cell Signaling, 2729, 1:500) was used. Complexes were pulled down with 20 µl prewashed Protein G Sepharose 4Fast Flow beads by incubation at 4°C for 3 hr. Three washes with wash buffer (50 mM Tris HCl, pH 8.0, 100 mM NaCl, 50 mM EDTA, pH 8.0, 0.1% IGEPAL CA-630, cOmplete ULTRA (Roche, 05892970001) and phosphatase inhibitor cocktail set V (EMD Millipore, 524629) was performed. Beads were boiled in 2x SDS-PAGE sample buffer and immunoprecipitation (IP) supernatants were analyzed by SDS-polyacrylamide gel electrophoresis. To confirm successful immunoprecipitation, PVDF membranes were incubated with rabbit anti-Src antibody (Cell Signaling, 2123, 1:1000). Src phosphorylation was analyzed by mouse anti-phosphotyrosine antibody (clone 4G10, Merck, 05–321, 1:1000). 1.5 µg proteins were loaded for input samples.

## Endothelial cell immunostaining

For immunostaining of HUVECs after knockdown experiments or Fc stimulation, cells were fixed with 4% PFA for 10 min. All the steps were performed at room temperature except otherwise mentioned. Cells were further incubated with 4% sucrose/PBS for 15 min, PBS rinsed and treated with 50 mM NH4Cl/PBS for 10 min. Cells were permeabilized with ice-cold 0.1% Triton X-100/PBS for 5 min at 4°C. After three PBS washes of 5 min, HUVECs were blocked with blocking buffer (4% donkey serum, 2% BSA in PBS) for 30 min. Primary antibodies were incubated in blocking buffer for 1 hr. After three PBS washes of 10 min, cells were incubated with DAPI (Sigma, D9542) and secondary antibodies for 30 min. Cells were PBS washed three times for 10 min and Fluoromount-G (Southern Biotech, 0100–01) was applied in each µ-Slide well (Ibidi, 80826). CD36 surface immunostaining was performed as described above without the cell permeabilization step.

Primary antibody used were: rabbit anti-CAV1 (Cell Signaling, 3238, 1:100), rabbit anti-pCAV1 (Y14) (Cell Signaling, 3251, 1:140), mouse anti-GM130 (BD Transduction Lab, 610822, 1:100), rabbit anti-GOLPH4 (Abcam, 28049, 1:500), goat anti-CDH5 (Biotechne, AF938, 1:70), mouse anti-CD36 (BD Biosciences, 552544, 1:100), Phalloidin Alexa Fluor-647 (Invitrogen, A22287, 1:40).

For analysis of EphB4 colocalization with pCAV1 (Y14), HUVECs were serum-starved overnight and stimulated with control human IgG/Fc (R and D Systems, 110-HG) or human ephrin-B2/Fc (Biotechne, 7397-EB) which were preclustered with goat anti-human IgG (Fc specific) (Jackson Immuno Research, 109-005-098, 10 µg/ml) for 30 min, followed by immunostaining, as described above. EphB4 immunosignal was detected with donkey anti-goat Alexa Fluor-488 (Invitrogen, A11055, 1:500).

For BSA, transferrin or BODIPY uptake experiments, HUVECs were washed twice before fixation with specific starvation medium. Immunostaining was performed as described above with minor modifications. Cells were fixed with 4% PFA for 10 min. After three PBS washes of 5 min, HUVECs were permeabilized with ice-cold 0.1% Triton X-100/PBS for 90 s on ice, with gentle shaking. Following three PBS washes of 5 min, cells were treated as described above.

HUVEC imaging was performed using Zeiss LSM780 and LSM880 confocal inverted microscopes. 10 random cells were selected per image and their shape was detected using VE-Cadherin junctional immunostaining. GM130 or GOLPH4 staining (depending on the type of experiment) were used to define the Golgi area within each cell.

Mean fluorescence intensity and normalized integrated density (Norm. IntDen) for CAV1, pCAV1 (Y14) or CD36 were calculated per whole-cell and Golgi area, using ImageJ version 2.0.0 (Fiji) (*Schindelin et al., 2012*). Mean Fluorescence Intensity (MFI) was calculated as the ratio of Norm. IntDen to the cell area.

## Fatty acid, BSA and Transferrin uptake assay

For BSA and transferrin uptake experiment HUVEC cells were serum starved in EGM-2 medium for 12 hr. For BODIPY uptake cells were serum starved in EGM2 medium containing 1% fatty-acids free BSA (Sigma-Aldrich, A9205) and 10 nM insulin (Sigma-Aldrich, I9278). Cells were incubated for 30 min with serum free medium containing BSA-555 (final concentration 4 µM, Molecular Probes, A34786) or with serum free medium containing 20 mM glucose and 1% BSA for Transferrin-488 (final concentration 50 µg/ml, Invitrogen, T13342) uptake. BODIPY $C_{12}$_500/510 (final concentration 5 µM,

Molecular Probes, D3823) uptake was performed in the above described medium. Stimulation with preclustered Fc proteins was performed simultaneously with the uptake molecule in the specific medium. After 30 minutes cells were washed with the uptake medium and were either fixed with 4% PFA for 10 min, immunostained and imaged, or directly prepared for FACS analysis. Cells were trypsinised (Sigma-Aldrich, T3924) and resuspended in FACS buffer (PBS containing 2% FCS and 2 mM EDTA). Cells were stained with DAPI (Sigma, D9542) for viability. The samples were measured using BD FACSVerse and analysed using FlowJo version 10.3.

### Stretching experiments

HUVECs were grown under the same conditions described above in Bioflex 6-well plates (Flex Cell, BF-3001U). Bioflex plates were pre-treated with 0.2% gelatin in PBS. HUVEC cells were transfected with *siEPHB4* and *siControl* as described above. 48 hr after siRNA transfection cells were stretched for 24 hr in Flexercell Strain Unit, FX-4000 Tension Plus (Flex Cell). Stretching conditions were 10% sinusoidal in cycles of 1 s.

Next, cells were fixed with 4% PFA for 15 min, washed twice with PBS for 5 min and permeabilized with PBS containing 0.1% Triton X-100 for 30 min. Cells where blocked for at least one hour with blocking solution (5% donkey serum, 3% BSA and 0.1% Triton X-100 in PBS). Primary antibodies were incubated in blocking solution overnight at 4°C. After 4 PBS washes of 5 min, secondary antibodies were incubated for 1 hr in PBS containing 5% BSA. Secondary antibodies were washed twice with PBS for 5 min and nuclei were contra-stained using DAPI (Sigma, D9542). Primary antibodies: mouse anti-Vinculin (Sigma, V9131, 1:100) and rabbit anti-CDH5 (Cell Signaling, 2500, 1:100). Cytoskeleton was contra-stained using Phalloidin Alexa Fluor-647 (Invitrogen, A22287, 1:100).

HUVECs imaging was performed using Leica SP8 confocal inverted microscopes. Disruption of the HUVEC cell monolayer was quantified by measuring the area devoid of cells after stretching. To do so, we measured the black pixels in the pictures using ImageJ version 2.0.0 (Fiji) (*Schindelin et al., 2012*). Vinculin and CDH5 colocalization in the junction area was measured by quantifying the number of voxels with both signals in Volocity software version 6.3 (Perkin Elmer).

### Atomic Force Microscopy

Force mapping was performed in EBM-2 endothelial cells medium (Lonza, CC-3156) supplemented with EGM-2 Single Quots (Lonza, CC-4176) stabilized with 10 mM HEPES (Sigma, H3537) at 37°C using a BioScope Catalyst-AFM (Bruker Nano Surfaces, Santa Barbara, California, USA) in closed-loop mode with a ramp size of 2 µm, max. force of 1nN and a tip velocity of 2.6 µm/s. A Large-Radius-Bio-Probe (Bruker AFM Probes, Camarillo, CA, USA; 'The nature of this technology is the subject of a non-provisional patent application to Bruker Nano, Inc currently pending at the United States Patent and Trademark Office.') (tip radius 3.46 µm) was used. Spring constant of the cantilever (0.176 N/m) was determined with an interferometer (OFV-551, Polytec, Waldbronn, Germany). Deflection sensitivity was adjusted according to the SNAP procedure (*Schillers et al., 2017*). Each force map contained 16 × 16 force-distance cycles over an area of 150 × 150 µm. The analysis of the force–indentation curves was performed with PUNIAS software (http://punias.free.fr/) using the linearized Hertz model (*Carl and Schillers, 2008*).

### Statistics

GraphPad's Prism software was used for statistical analysis of all experiments but RNA-seq and MS. Unpaired two-tailed *t* test with Welch's correction was used for comparison between two groups. For multiple comparisons, one-way ANOVA with Sidak's multiple comparisons test was used. Data is presented as scatter plots with mean ± standard error of mean (s.e.m). Differences were considered statistically significant at $p < 0.05$.

## Acknowledgements

We thank T Mäkinen and F Kiefer for *Prox1^CreERT2* mice, C Bätza, R Holtmeier, A Blaque, A Büchsenschütz, and K Mildner for excellent technical assistance. We thank R Unger for the Flexercell Strain Unit. GL would like to thank the CiM young academy, and especially S Eligehausen, for the support and constructive discussions and ideas. We are grateful to R Diéguez-Hurtado for critically reading the manuscript and to B Hernández-Rodríguez for assistance with RNA-seq analysis reproducibility.

This work was funded by the DFG (CRC 1348), the Deutsche Forschungsgemeinschaft Cells-in-Motion Cluster of Excellence (EXC 1003–CiM) through the Pilot Project PP-2015–12 to GL and ND, and the Max Planck Society. GL was supported by an EMBO postdoctoral fellowship (ALTF 421–2014).

## Additional information

### Competing interests

John Wiseman: is affiliated with AstraZeneca. The author has no financial interests to declare. The other authors declare that no competing interests exist.

### Funding

| Funder | Grant reference number | Author |
|---|---|---|
| Max-Planck-Gesellschaft | Open-access funding (CRC 1348) | Ralf H Adams |
| Deutsche Forschungsgemeinschaft | SFB 1348 and Cells-in-Motion Cluster of Excellence (EXC 1003-CiM) | Mara E Pitulescu Ralf H Adams |
| Deutsche Forschungsgemeinschaft | Cells-in-Motion Cluster of Excellence (EXC 1003–CiM) – Pilot Project PP-2015–12 | Guillermo Luxán Noelia Díaz |
| EMBO | Postdoctoral fellowship – ALTF 421–2014 | Guillermo Luxán |

The funders had no role in study design, data collection and interpretation, or the decision to submit the work for publication.

### Author contributions

Guillermo Luxán, Conceptualization, Data curation, Formal analysis, Investigation, Writing—original draft; Jonas Stewen, Formal analysis, Validation, Investigation; Noelia Díaz, Data curation, Formal analysis, Investigation, Methodology; Katsuhiro Kato, Sathish K Maney, Anusha Aravamudhan, Nina Nagelmann, Hannes CA Drexler, Dagmar Zeuschner, Sven Hermann, Investigation, Methodology; Frank Berkenfeld, Formal analysis, Investigation, Methodology; Cornelius Faber, Investigation, Methodology, Project administration; Hermann Schillers, Supervision, Investigation, Methodology; John Wiseman, Resources, Investigation, Methodology; Juan M Vaquerizas, Data curation, Formal analysis, Supervision; Mara E Pitulescu, Conceptualization, Data curation, Supervision, Investigation, Methodology, Writing—original draft, Writing—review and editing; Ralf H Adams, Conceptualization, Supervision, Funding acquisition, Writing—original draft, Project administration

### Author ORCIDs

Guillermo Luxán [ID] https://orcid.org/0000-0001-8350-8659
Noelia Díaz [ID] http://orcid.org/0000-0002-0319-3448
Cornelius Faber [ID] http://orcid.org/0000-0001-7683-7710
Juan M Vaquerizas [ID] https://orcid.org/0000-0002-6583-6541
Ralf H Adams [ID] https://orcid.org/0000-0003-3031-7677

### Ethics

Animal experimentation: All animal procedures were performed according to relevant laws and institutional guidelines, were approved by local animal ethics committees and were conducted with permissions (84-02.04.2015.A185, 84-02.04.2017.A238, 81-02.04.2019.A114) granted by the Landesamt für Natur, Umwelt und Verbraucherschutz (LANUV) of North Rhine-Westphalia.

### Decision letter and Author response

Decision letter https://doi.org/10.7554/eLife.45863.sa1

Author response https://doi.org/10.7554/eLife.45863.sa2

## Additional files

### Supplementary files
• Supplementary file 1. Key Resources Table.List of all the materials and resources used.

• Transparent reporting form

### Data availability
All mass spectrometry proteomics data have been deposited to the ProteomeXchange Consortium (http://proteomecentral.proteomexchange.org) via the PRIDE partner repository: Dataset identifier PXD012575. RNA-seq data have been deposited in the ArrayExpress database at EMBL-EBI (www.ebi.ac.uk/arrayexpress): Project E-MTAB-7686.

The following datasets were generated:

| Author(s) | Year | Dataset title | Dataset URL | Database and Identifier |
|---|---|---|---|---|
| Guillermo Luxán, Hannes CA Drexler | 2019 | Endothelial EphB4 maintains vascular integrity and transport function in adult heart | https://www.ebi.ac.uk/pride/archive/projects/PXD012575 | PRIDE, PXD012575 |
| Noelia Díaz, Guillermo Luxán | 2017 | RNA-seq of 12-weeks-old male murine heart ventricles | https://www.ebi.ac.uk/arrayexpress/experiments/E-MTAB-7686/ | ArrayExpress, E-MTAB-7686 |

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
