## [Decision Letter]

**Acceptance summary:**

This is now a comprehensive study linking anatomical and functional defects in the coronary vasculature after endothelial-specific knockout of EphB4, with a molecular pathway for lipid uptake though a caveolin-mediated pathway.

**Decision letter after peer review:**

Thank you for submitting your article "Endothelial EphB4 maintains vascular integrity and transport function in adult heart" for consideration by *eLife*. Your article has been reviewed by three peer reviewers, one of whom is a member of our Board of Reviewing Editors, and the evaluation has been overseen by Harry Dietz as the Senior Editor. The reviewers have opted to remain anonymous.

The reviewers have discussed the reviews with one another and the Reviewing Editor has drafted this decision to help you prepare a revised submission.

Summary:

This is an interesting and potentially important study showing that EC-specific deletion of EphB4 leads to coronary EC branching defects, pericyte coverage, microhemorrhage, and as a result increased heart weight/tibial length, hypertrophic CMs, reduced LV chamber mass and reduced diastolic and systolic function. RNAseq and proteomics analysis and follow up studies in HUVECs and mice suggest metabolic dysfunction, specifically defects in lipid uptake via disruption of caveolin (*CAV1*) levels and trafficking.

Overall the narrative is credible. However, as the story develops around lipid uptake and caveolae dysfunction, the links between in vitro and in vivo data become uncomfortably fragmented. Reviewers identified issues surrounding the relevance of the HUVEC model, technical aspects of siRNA experiments, quantification of data, and, in particular, the missed opportunity to strengthen links between in vitro data and the mutant mouse phenotype. Since HUVECs do not exhibit the properties of heart capillary ECs, the results cannot be used to make conclusions about heart ECs without strong supporting evidence (a key theme of the paper is the difference between vascular beds). We invite the authors to address the reviewers' comments with these criticisms in mind and specifically to provide additional data that strengthen the links between in vitro and in vivo data.

Essential revisions:

1) TEMs are hard for the novice to interpret. They need better labels for cell types, including blood cells, and structures. Legends to Figure 2D and Figure 2—figure supplement 3 are vague. A better connection between TEM and phenotypes in vitro needs to be made (see below). It would be helpful if the differences in the TEMs were quantified. It appears that there are still many vesicles in some regions of the mutant shown on the left in Figure 2D; however, the panels shown for control do not allow the distribution to be compared.

2) Is fibrosis a component of the cardiac condition in cKO mice? This could impact biomechanics and vessel dysfunction and fragility.

3) Figure 4—figure supplement 2A is very unclear. What is the specific point here? VEGFR3 staining represents lymphatics. Green EphB4 is hardly visible so the impact of KO is not discernible. Why aren't the arrows pointing to VEGFR3+ structures?

4) The authors used whole heart ventricle tissue for RNA-sequence and proteomics study. Without functional studies, it is difficult to assess whether EphB4 deficiency in ECs leads to dysregulation in capillary ECs, cardiomyocytes, or in both. Authors make several unvalidated assumptions based on RNA-sequence and proteomic analysis. These should be qualified.

5) Subsection “EphB4 controls fatty acid uptake and focal adhesion in cultured ECs”, second paragraph: EphB4 signaling controls PI3K and MAPK/ERK signaling and these are slightly reduced in KD HUVEC cells. However, it is unclear if this is direct or indirect. How could the authors better connect this impact with effects on *CAV1* and caveoli function? What is the biochemical impact of PI3K and AKT and ERK on *CAV1*? Can this be studied?

6) The reduction of BSA uptake by KD HUVEC cells is slight (~5%) – in this sense it is not believable to show the fluorescence figure apparently indicating stronger reduction in uptake. Is this result credible? This seems to need stronger support.

7) Figure 6C. The effect is significant but very small. Did the author knockdown a positive control gene to show the magnitude of the difference with depletion of a gene known to regulate uptake? Or alternatively, cite another paper with such analysis. This would provide confidence that the small difference with EphB4 deletion is biologically significant.

8) Figure 6E. Although levels of *CAV1* seem to be reduced, one could argue that the strong signal in 2 of the 3 control cells shown are also peri-nuclear. This is also true of cells treated with ephrin B2-Fc. Without further quantification of intracellular compartmentalization, this is insufficient evidence for disruption of *CAV1* distribution. How does this data relate to TEM showing apparent build-up of caveolae in ECs. More data is required on localisation of *CAV1* in vitro and caveolae in vivo.

9) The rationale for studying the role of EphB4 in regulating caveolar function in heart capillary endothelium is unclear. Src-dependent phosphorylation of caveolin-1 (Cav-1) on Y-14 is implicated in macromolecular cargo endocytosis. Did authors observe dysregulation of Cav-1 phosphorylation in ECs of EphB4^ΔEC^ mice? If so, does Src activation in heart capillary ECs require EphB4? Does EphB4 physically associated with Cav-1 in caveolar microdomains?

10) Other experiments using immunofluorescence are sometimes missing quantification, such as in Figure 7C and D and other panels. Is the difference shown in representative images statistically different among many cells when quantified?

11) A general but potentially important issue relating to siRNA reagents is revealed in Supplementary Figure 8-9, which show the flow plots and gating strategies. These vary considerably wrt the SSC-A/DAPI plots between experiments (Supplementary Figures 8B and 9A) and in some cases between *siControl* and siEphB4, where there seems to be an increase in DAPI+ cells in siEphB4 sample, suggesting increased cell death. The plot is very different when cells are treated with Fc or ephrin B2-Fc. What are the implications of this variability – perhaps the changes in BSA uptake are related to the differences in cell death or stress in the culture?

12) The CD36 data is gleaned from IF only. This should be quantified by flow. The same for Figure 7D. If robust, the latter presents a good assay for how EphrinB2/EphB4 controls lipid uptake via caveolin. Could this be exploited?

13) The authors comment in the Discussion that EphB4 associates with *Cav1* and induces phosphorylation of Tyr-14. Can this be exploited in the current study?

14) Have the authors considered a more direct test of their model, such as directly measuring fatty acid uptake in the heart in mutants or rescuing with CD36 re-expression?

15) Did authors observe any vascular abnormalities in other highly vascularized organ such as lung and liver?

[Editors' note: further revisions were requested prior to acceptance, as described below.]

Thank you for submitting your revision for the article "Endothelial EphB4 maintains vascular integrity and transport function in adult heart" for consideration by *eLife*. Your article has been reviewed by the Reviewing Editor and overseen by Harry Dietz as the Senior Editor. No reviewers were consulted for this submission.

The narrative of the manuscript is very much improved with the addition of extensive new data and quantifications. Before acceptance, the authors should address the following points.

1) please add arrows indicating intra-mitochondrial glycogen granules in cardiomyocytes in Figures 2D and Figure 2—figure supplement 3A-D.

2) Please clarify in the context of the subsection “Loss of endothelial EphB4 causes heart hypertrophy” and Figure 1—figure supplement 1A-C, the overall expression patterns of EphB4 and EphrinB2. The immunofluorescence and text suggest highest staining in large veins for EphB4 but not in capillaries or arteries, whereas the efnb3-GFP is expressed in arteries and capillaries. If it is believed that EphB4 is expressed only in the venous coronary system, in the Discussion, the authors should return to this fact in light of the phenotypes discovered in vivo and the known biology around fatty acid metabolism, role of *Cav1* in endothelial cell integrity etc. This issue seems to have been glossed over in the manuscript and discussions (albeit that HUVECs were used as in vitro model). Is it possible that EphB4 is indeed expressed at lower levels in capillaries.

3) Quantifications of pCav, *Cav1* after knockdown (KD) of EphB4 are confusing. Western blots show both *Cav1* and pCav1 are reduced after KD. Immunostainings suggest that EphB4 KD reduces *Cav1* and pCAV1 in the cytoplasm (Figure 7D), yet the quantifications do not show this. The issue of their expression in golgi is impossible to assess as the images do not show sufficient detail. Do the authors stand by this data, and if so please explain the disconnect between the western blots and immunostainings, and the quantification of expression in cell compartments in Figure 7A, D, E. Please show in revised manuscript examples of altered expression of golgi.

4) In Figure 8A, treatment of HUVEC cells with B2/Fc induced pCav1 but not total *Cav1*. This is inhibited by PP2 (inhibitor for Src signalling). Figure 8B, C shows quantifications making the point that B2/Fc stimulate both pCav1 and pCav1/*Cav1*, and that increases in both are inhibited in the presence of PP2. Figure 8C shows the PP2 effect on pCav1 in cells. Figure 8D show that the increase in pCav1/*Cav1* with B2/Fc is not inhibited by Ly294002 and U0126 (inhibitors of PI3K and AKT). However, in Figure 8E, quantification of the increase in pCav1 in the cytoplasm versus golgi of HUVEC cells with B2/Fc, as judged by immunostaining, are shown in the presence of DMSO and inhibitors. Here, the PP2 inhibition is confirmed (PP2 also inhibits the basal level of pCav1 without B2/Fc stimulation. But LY294002 and U0126 also inhibit the B2/Fc-induced increase in pCav1, which seems to conflict with Figure 8D. There is apparently no impact on golgi localisation of pCav1. Please explain.

---

## [Author Response]

[…] Overall the narrative is credible. However, as the story develops around lipid uptake and caveolae dysfunction, the links between in vitro and in vivo data become uncomfortably fragmented. Reviewers identified issues surrounding the relevance of the HUVEC model, technical aspects of siRNA experiments, quantification of data, and, in particular, the missed opportunity to strengthen links between in vitro data and the mutant mouse phenotype. Since HUVECs do not exhibit the properties of heart capillary ECs, the results cannot be used to make conclusions about heart ECs without strong supporting evidence (a key theme of the paper is the difference between vascular beds). We invite the authors to address the reviewers' comments with these criticisms in mind and specifically to provide additional data that strengthen the links between in vitro and in vivo data.

We have provided in the revised version all the details relevant for the knockdown experiments and the quantification methods. We used HUVECs to study the mechanism, a cell line widely used for the in vitro studies on endothelial cells.

Essential revisions:1) TEMs are hard for the novice to interpret. They need better labels for cell types, including blood cells, and structures. Legends to Figure 2D and Figure 2—figure supplement 3 are vague. A better connection between TEM and phenotypes in vitro needs to be made (see below). It would be helpful if the differences in the TEMs were quantified. It appears that there are still many vesicles in some regions of the mutant shown on the left in Figure 2D; however, the panels shown for control do not allow the distribution to be compared.

We agree. As suggested by the reviewers, we have now included labels for different cell types in Figure 2D and Figure 2—figure supplement 3.We have also quantified the total number of caveolar vesicles per capillary and the number of vesicles associated to the basolateral membrane. Quantification is now shown in Figure 2—figure supplement 3C, which also confirms that there are strong and statistically significant differences. The inclusion of the graphs will be greatly helpful for readers that are not familiar with electron micrographs or subcellular structures.

Accordingly, we have now rewritten the relevant sections of the figure legends:

“Figure 2. (D) Electron micrographs of control and *Ephb4*^∆EC^ capillaries. Bottom images are higher magnifications of boxed areas in upper panels. Arrowheads indicate accumulation of caveolar vesicles at the mutant endothelial basolateral membrane (center) and thrombocytes in a vascular rupture (right). Erythrocytes (er), thrombocytes (th), cardiomyocytes (cm) and endothelial cells (ec) are indicated.”

“Figure 2—figure supplement 3. Ultrastructural analysis of *Ephb4* mutant hearts.

(A-C) Electron micrographs show details of the ventricular wall of the left ventricle. (A) Mutant endothelial cells are thickened and present irregular lining (arrowheads). […] (D) Cardiomyocytes in mutant hearts show accumulation of glycogen (arrowheads) close to mitochondria. Erythrocytes (er), thrombocytes (th), cardiomyocytes (cm) and endothelial cells (ec), and pericytes (pc) are indicated.”

2) Is fibrosis a component of the cardiac condition in cKO mice? This could impact biomechanics and vessel dysfunction and fragility.

To assess this important question, we have performed immunostainings for collagen I and IV as well as Masson Trichrome staining of adult control and *Ephb4*^∆EC^ heart sections. These experiments did not reveal any appreciable signs of fibrosis in the mutant myocardium. The results are presented in Figure 2—figure supplement 2and mentioned in the text as follows: “It is also noteworthy that loss of EphB4 in ECs did not induce fibrosis in the mutant myocardium (Figure 2—figure supplement 2A, B)”.

“Figure 2—figure supplement 2. *Ephb4* inactivation does not induce fibrosis in heart. (A) Immunohistochemistry on for Collagen I (ColI, green) and Collagen IV (ColIV, green) in transverse sections of control and *Ephb4*^∆EC^ heart showing both ventricles. […] No fibrosis is observed. lv, left ventricle, rv, right ventricle.”

3) Figure 4—figure supplement 2A is very unclear. What is the specific point here? VEGFR3 staining represents lymphatics. Green EphB4 is hardly visible so the impact of KO is not discernible. Why aren't the arrows pointing to VEGFR3+ structures?

We thank the reviewers for this comment and we agree that data presentation in this figure was clearly suboptimal. The *Cdh5-CreERT2* transgenic mice used in our study are active in the endothelial cells of both blood vessels and the lymphatic vasculature. Given that both vessel types are important for cardiac function, we selectively inactivated *Ephb4* in lymphatic ECs with the help of the *Prox1-CreERT2* allele. The resulting *Ephb4*^∆LEC^ mutants retain EphB4 immunostaining in veins (indicated by arrows) but lost the signal in VEGFR3+ lymphatic vessels (indicated by arrowheads; see Figure 4—figure supplement 2). The images, data presentation and labeling have been improved in the revised manuscript and it is now much easier to understand what is shown in the figure.

4) The authors used whole heart ventricle tissue for RNA-sequence and proteomics study. Without functional studies, it is difficult to assess whether EphB4 deficiency in ECs leads to dysregulation in capillary ECs, cardiomyocytes, or in both. Authors make several unvalidated assumptions based on RNA-sequence and proteomic analysis. These should be qualified.

Our RNA-seq and proteomics experiments were performed using whole heart ventricles aiming to identify changes that would help us to interpret the phenotypic and functional changes of heart function, identified by echocardiography, CMRI analysis and TEM (Figure 1 and 2).

5) Subsection “EphB4 controls fatty acid uptake and focal adhesion in cultured ECs”, second paragraph: EphB4 signaling controls PI3K and MAPK/ERK signaling and these are slightly reduced in KD HUVEC cells. However, it is unclear if this is direct or indirect. How could the authors better connect this impact with effects on CAV1 and caveoli function? What is the biochemical impact of PI3K and AKT and ERK on CAV1? Can this be studied?

Agree. The in vitro part of the manuscript has been revised extensively and extended (see Figures 6-8 and 9A-C). Data presented in the original Figures 6 and 7 has been replaced or moved to the supplementary figures. In the first set of new experiments, we stimulated HUVECs with recombinant, soluble proteins (namely ephrin-B2/Fc to stimulate EphB4, EphB4/Fc to trigger reverse signaling through ephrin-B2, and IgG/Fc as control) and analyzed the activation of different downstream signaling pathways. Concurrent treatment with inhibitors of Akt (LY294002), MEK (U0126), and Src (PP2) was used to validate the specificity of the observed effects. The resulting data indicate that EphB4 activation with ephrin-B2/Fc leads to the activation of Akt and Src, whereas ERK is strongly reduced (new Figure 6A, B). We have also included new data showing that Src phosphorylation is significantly reduced after *EPHB4* knockdown in HUVECs (new Figure 7). In the rest of the study, we focus mainly on Src signaling because this pathway plays a critical role in caveolar function.

Next, we investigated how EphB4 signaling affects caveolin 1 (*CAV1*) function. Here we concentrated on *CAV1* phosphorylation on tyrosine 14 (Y14), which is known to regulate caveolar trafficking, and *CAV1* localization. New Western blots prove that ephrin-B2/Fc-mediated activation of EphB4 induce an increase in phosphorylated *CAV1* (pCAV1) total protein level (new Figure 8A, B), which is absent after knockdown of *EPHB4* (new Figure 8F). Furthermore, ephrin-B2/Fc-induced *CAV1* phosphorylation is reduced by the Src inhibitor PP2 but not by U0126 or LY294002 treatment (new Figure 8B, D). pCAV1 is also reduced in *siEPHB4*-treated HUVECs or after knockdown of *CAV1*, which we have used as a positive control (new Figure 7A, C).

In another set of experiments, we show that Src inhibition reduces the levels of cytoplasmic pCAV1 but not the Golgi localized pool both without stimulation (control-Fc treatment) and after ephrin-B2/Fc treatment (new Figure 8C, E) (see also answer to question 8). In contrast, inhibition of Akt or ERK did not substantially affect *CAV1* or pCAV1 cytoplasmic and Golgi fractions.

In summary, our data establish that signaling by ephrin-B2 and EphB4 controls *CAV1* phosphorylation through Src activation.

6) The reduction of BSA uptake by KD HUVEC cells is slight (~5%) – in this sense it is not believable to show the fluorescence figure apparently indicating stronger reduction in uptake. Is this result credible? This seems to need stronger support.

We agree with the reviewers that the flow cytometric analysis of BSA uptake revealed only a small reduction (~5%) upon *EPHB4* KD (these data have been relocated toFigure 9—figure supplement 1). Therefore, we included quantitation of fluorescently labeled BSA per cell (3 images, 10 cells randomly selected were quantified per image). Cell shape was defined using VE-Cadherin/CDH5 junctional labeling. This approach showed a higher decrease (~30%) in BSA uptake (as shown in new Figure 9 A, C),which is comparable to the effect seen after *CAV1* knockdown (see answer to question 7).Consistent with our previous results (see Figure 9—figure supplement 1C), transferrin uptake was not affected in *EPHB4* knockdown cells (see new data in Figure 9—figure supplement 1E, F).

7) Figure 6C. The effect is significant but very small. Did the author knockdown a positive control gene to show the magnitude of the difference with depletion of a gene known to regulate uptake? Or alternatively, cite another paper with such analysis. This would provide confidence that the small difference with EphB4 deletion is biologically significant.

As suggested by the reviewers to compare the extent of albumin uptake impairment upon *EPHB4* depletion with depletion of a gene known to regulate BSA uptake, we interfered with *CAV1* expression in HUVECs. Several studies have shown a critical role for *CAV1* and caveolae in mediating albumin transcytosis in endothelial cells (e.g. Simionescu N., Int. J Cell Biol., 1981; Schubert W. et al., JBC, 2001; Ghitescu L. et al., JCB, 1986).

Our results (shown in the new Figure 9A, C) indicate that BSA uptake is similarly reduced in *EPHB4* KD and *CAV1* KD HUVECs. The quantitation graphs compare normalized integrated density per cell in control versus knockdown cells.

It was previously shown that the transcellular transport of albumin is not only mediated via caveolae, but there are also *CAV1*-independent pathways (e.g. Predescu et al., Am. J. Physiol Lung Cell Mol Physiol, 2007; Li H-H. et al., 2013). This, along with incomplete depletion of *CAV1* protein in *CAV1* KD cells, might explain the residual BSA uptake seen in our experiments.

8) Figure 6E. Although levels of CAV1 seem to be reduced, one could argue that the strong signal in 2 of the 3 control cells shown are also peri-nuclear. This is also true of cells treated with ephrin B2-Fc. Without further quantification of intracellular compartmentalization, this is insufficient evidence for disruption of CAV1 distribution. How does this data relate to TEM showing apparent build-up of caveolae in ECs. More data is required on localisation of CAV1 in vitro and caveolae in vivo.

We followed the reviewer’s suggestion to investigate *CAV1* intracellular compartmentalization. Indeed, we observed that *CAV1* and pCAV1 partially co-localize with Golgi markers (GM130 and GOLPH4) (new Figure 7D and 8C). Therefore, we quantified *CAV1* and pCAV1 immunosignal in Golgi and cytoplasm (excluding the intensities localized to Golgi and nuclear areas) in knockdown experiments and upon Fc protein stimulation (control Fc, ephrin-B2/Fc and EphB4/Fc). The resulting data indicate that the total level of *CAV1* and pCAV1 (as quantified by Western blotting from total cell lysates) as well as immunosignals in the Golgi area of *EPHB4* KD cells are reduced, whereas the cytoplasmic fraction is not affected. These results indicate a change in intracellular distribution of *CAV1* and pCAV1 associated with possible reduced synthesis or transport to the Golgi. These results also match well to the in vivo TEM data, which show an increased number of basolateral caveolae in *Ephb4* mutant capillary endothelium, arguing for impaired trafficking of caveolae.

We also show that ephrin-B2/Fc stimulation increases pCAV1 in total protein lysate (shown by Western blotting) and its cytoplasmic localization but not the Golgi localized fraction (new Figure 8C, E). Furthermore, the increase in phosphorylated *CAV1* protein after ephrin-B2/Fc stimulation is reduced by knockdown of *EPHB4* (new Figure 8F).

Altogether, these data indicate that EphB4 forward signaling is necessary for *CAV1* localization and phosphorylation.

9) The rationale for studying the role of EphB4 in regulating caveolar function in heart capillary endothelium is unclear. Src-dependent phosphorylation of caveolin-1 (Cav-1) on Y-14 is implicated in macromolecular cargo endocytosis. Did authors observe dysregulation of Cav-1 phosphorylation in ECs of EphB4ΔEC mice? If so, does Src activation in heart capillary ECs require EphB4? Does EphB4 physically associated with Cav-1 in caveolar microdomains?

Our proteomic analysis revealed a downregulation of proteins involved in endocytic recycling in *Ephb4* mutant heart ventricles (Figure 5—figure supplement 1G) and TEM images showed that mutant ECs had an expanded cytoplasm with few vesicles, whereas caveolar vesicles accumulate along the basolateral membrane (new Figure 2D and Figure 2—figure supplement 3A-D). Prompted by these findings, we decided to investigate the link between EphB4 and *CAV1*, one of the main components of caveolae. Our in vitro data (see detailed answers to questions 5-8 above) indicate that EphB4 regulates caveolar function in a Src and *CAV1*-dependent fashion.

While we do not have direct data on Src phosphorylation in cardiac ECs in vivo, our findings on the link between EphB4, Src and *CAV1* are consistent with recent preprint showing that EphB1, a related Eph receptor, binds the *CAV1* scaffold domain in cultured ECs (Tiruppathy C. et al., bioRxiv, 2019). EphB1 activation by ephrin-B1 resulted in uncoupling of EphB1 from *CAV1*, which allowed autophosphorylation of EphB1 on Y600, followed by Src binding to EphB1. This resulted in Src activation with further downstream phosphorylation of *Cav1* on Y14. Furthermore, the authors demonstrated that EphB1 / *Cav1* interaction was required for signaling caveolae-mediated endocytosis and deletion of *Ephb1* in mice severely depleted caveolae numbers. Consequently, the internalization of Alexa Fluor-594 labeled bovine serum albumin tracer (Alexa-594 BSA) was 4-fold reduced in *EphB1*^−/−^-EC compared to WT ECs. Based on previous work, EphB1 and the ligand ephrin-B1 are, unlike ephrin-B2 and EphB4, not important for the regulation of EC function in vivo. Nevertheless, the mechanistic findings in cultured cells are consistent in both studies.

To investigate if EphB4 and *CAV1* colocalize in microdomains, we stimulated cells with control Fc or ephrin-B2/Fc chimeric proteins. After 30 minutes of stimulation with ephrin-B2/Fc, EphB4 partially co-localizes with pCAV1 in intracytoplasmic microdomains and at cellular junctions (new Figure 6C). These results are similar to the data obtained by Muto and colleagues who showed that stimulated EphB4 associates with *CAV1* (Muto et al., 2011). In the current study, we did not perform experiments to prove that EphB4 needs to physically interact with *CAV1* and, instead, it is entirely feasible that EphB4-dependent Src activation upregulates pCAV1 without direct binding between EphB4 and *CAV1*.

Despite of these open questions, which will be surely the subject of future studies, our data link EphB4 signaling to caveolar function and, as we show in Figure 9A-Cthe uptake of albumin and fatty acids by ECs.

10) Other experiments using immunofluorescence are sometimes missing quantification, such as in Figure 7C and D and other panels. Is the difference shown in representative images statistically different among many cells when quantified?

We agree and we performed detailed quantitation (as shown in Figure 7, 8, 9; Figure 8—figure supplement 1 and Figure 9—figure supplement 1) and present the statistical significance in the figures and source file). Figure 7C and D from the original submission were replaced by new data shown in Figure 9 and Figure 9—figure supplement 1.

11) A general but potentially important issue relating to siRNA reagents is revealed in Supplementary Figure 8-9, which show the flow plots and gating strategies. These vary considerably wrt the SSC-A/DAPI plots between experiments (Supplementary Figures 8C and 9A) and in some cases between siControl and siEphB4, where there seems to be an increase in DAPI+ cells in siEphB4 sample, suggesting increased cell death. The plot is very different when cells are treated with Fc or ephrin B2-Fc. What are the implications of this variability – perhaps the changes in BSA uptake are related to the differences in cell death or stress in the culture?

We have grouped the flow plots and gating strategies from all uptake experiments into the new Figure 9—figure supplement 1. We understand the reviewer’s concern, but there are no experimental differences between the *siControl* and *siEPHB4* treated cells within the same type of experiment. However, the SSC-A / DAPI flow plots are different due to the fact that experiments were not all performed simultaneously and cells were exposed to different tracers (BSA-555 / Transferrin-488 / BODIPY C_12_-500/510), which involved different compensation setups and gating strategies for different fluorophores (PE / FITC) in combination with DAPI gating.

12) The CD36 data is gleaned from IF only. This should be quantified by flow. The same for Figure 7D. If robust, the latter presents a good assay for how EphrinB2/EphB4 controls lipid uptake via caveolin. Could this be exploited?

We have now quantified the immunofluorescence signal of CD36 in cytoplasm and Golgi per cell (new Figure 7B-E, 8G and Figure 8—figure supplement 1) and have evaluated CD36 levels as well as its change in localization upon different treatments (knockdown of *EPHB4* or *CAV1* or Fc stimulation +/- inhibitors treatments). We found no evidence for co-localization of EphB4 and CD36 (data not shown) but knockdown of *EPHB4* or *CAV1* and PP2 treatment affect its subcellular distribution of the fatty acid translocase in vitro. Nevertheless, our data indicate that fatty acid uptake is strongly reduced upon *EPHB4* (or *CAV1*) knockdown, which is consistent with the phenotypic alterations in *Ephb4* mutant hearts in vivo.

in vivo, the effect of *Ephb4* loss-of-function on CD36 is more dramatic andresults in decreased CD36 in ECs and increased CD36 levels in cardiomyocytes, probably as a compensatory response (new Figure 9). In conjunction with this, RNA-seq and proteomics data point to a role of EphB4 regulating heart metabolism. Thus, RNA-seq analysis revealed that genes related to the regulation of carbohydrate metabolism and to the positive regulation of glycolysis were upregulated in mutant samples. In addition, MS analysis of heart ventricles indicated un upregulation of fatty acid catabolic processes accompanied by a downregulation of endocytic recycling. These data indicate that *Ephb4* mutants have defects in transport processes and a metabolic shift towards glycolysis in heart.

13) The authors comment in the Discussion that EphB4 associates with Cav1 and induces phosphorylation of Tyr-14. Can this be exploited in the current study?

See our response to Essential revision #9.

14) Have the authors considered a more direct test of their model, such as directly measuring fatty acid uptake in the heart in mutants or rescuing with CD36 re-expression?

We have considered this experiment, but, unfortunately, we were not able to directly measure fatty acid uptake in vivodue to the fact that carbon-11-labeled tracers have a very short half-life of 20 minutes.

15) Did authors observe any vascular abnormalities in other highly vascularized organ such as lung and liver?

We have performed immunostaining of liver and kidney sections of control and *Ephb4*^∆EC^mice and could not observe any overt changes (see new Figure 4—figure supplement 1).

[Editors' note: further revisions were requested prior to acceptance, as described below.]The narrative of the manuscript is very much improved with the addition of extensive new data and quantifications. Before acceptance, the authors should address the following points.1) please add arrows indicating intra-mitochondrial glycogen granules in cardiomyocytes in Figures 2D and Figure 2—figure supplement 3A-D.

We have marked by arrowheads the intra-mitochondrial glycogen granules in cardiomyocytes in Figure 2D (yellow arrowheads) and Figure 2—figure supplement 3D (white arrowheads).

2) Please clarify in the context of the subsection “Loss of endothelial EphB4 causes heart hypertrophy” and Figure 1—figure supplement 1A-C, the overall expression patterns of EphB4 and EphrinB2. The immunofluorescence and text suggest highest staining in large veins for EphB4 but not in capillaries or arteries, whereas the efnb3-GFP is expressed in arteries and capillaries.

We have described EphB4 and ephrin-B2 expression pattern in the main text as shown below:

“The receptor tyrosine kinase EphB4 and its ligand ephrin-B2 are expressed in the capillary plexus of the adult coronary vasculature. In addition, EphB4 is expressed in large veins, whereas ephrin-B2 is restricted to arteries (Figure 1—figure supplement 1A, B).”

We have marked EphB4 expression in mouse cardiac capillaries with blue arrowheads in Figure 1—figure supplement A, D. Expression of EphB4 in capillaries appears indeed much lower than in veins. Moreover, EphB4 expression might seem lower than ephrin-B2, which mainly reflects that we were detecting the latter using *Efnb2::GFP* reporter mice. This results in strong nuclear GFP signal and is therefore a very sensitive method for the detection of gene expression at the *Efnb2* locus.

If it is believed that EphB4 is expressed only in the venous coronary system, in the Discussion, the authors should return to this fact in light of the phenotypes discovered in vivo and the known biology around fatty acid metabolism, role of Cav1 in endothelial cell integrity etc. This issue seems to have been glossed over in the manuscript and discussions (albeit that HUVECs were used as in vitro model). Is it possible that EphB4 is indeed expressed at lower levels in capillaries.

We are grateful for this comment and the opportunity to clarify a few important issues. While ephrin-B2 and EphB4 are frequently used as arterial and venous markers, respectively, research done by us and others indicated that both molecules are also present in the capillary endothelium and, in fact, this might be the site where these molecules interact and control EC behavior (Sawamiphak et al., 2010; Wang et al., 2010; Groppa et al. 2018, EMBO Rep. 19: e45054). This is also supported by staining with very sensitive (LacZ) reporter alleles both for ephrin-B2 and EphB4.

Based on your comments, we have improved our discussion of EphB4 and ephrin-B2 expression, EphB4-mediated regulation of *CAV1*, and CD36 regulated fatty acid transport from endothelial cells to cardiomyocytes. A version of the article with tracked changes is submitted together with the revised file.

3) Quantifications of pCav, Cav1 after knockdown (KD) of EphB4 are confusing. Western blots show both Cav1 and pCav1 are reduced after KD. Immunostainings suggest that EphB4 KD reduces Cav1 and pCAV1 in the cytoplasm (Figure 7D), yet the quantifications do not show this. The issue of their expression in golgi is impossible to assess as the images do not show sufficient detail. Do the authors stand by this data, and if so please explain the disconnect between the western blots and immunostainings, and the quantification of expression in cell compartments in Figure 7A, D, E. Please show in revised manuscript examples of altered expression of golgi.

We are grateful for this comment and we agree that the differences seen in the immunostaining experiments were not properly reflected by the quantitative analysis. Moreover, a careful analysis of pCAV1 distribution indicated that only a minor fraction (0.8% in average of the whole cell immunosignal) of the immunosignal overlaps with a Golgi marker. Furthermore, we realized that both the knockdown of *EPHB4* and *CAV1* lead to substantially increased cell size (visualized by anti-VE-Cadherin/CDH5 staining; see Figure 8—figure supplement 1A), which needs to be taken into account in our quantitation.

We have therefore reanalyzed all the relevant data, removed the graphs for IF signal in cytoplasm and Golgi, and now show mean fluorescent intensity (MFI) of pCAV1 for *EPHB4* and *CAV1* knockdown experiments (Figure 7E; Figure 9C). This approach confirms the reductions in *CAV1* and pCAV1 signals seen in *siEPHB4* and *siCAV1* treated HUVECs (Figure 7D). The relevant figures and raw data files have been updated accordingly.

The new quantitations of pCAV1 immunosignals are also consistent with the Western blotting results (Figure 7A-C) where equal amounts of protein lysate were compared between *EPHB4* KD, *CAV1* KD and control HUVECs, and the amounts of *CAV1*, pCAV1 and CD36 were normalized to GAPDH internal control.

In the quantitation of all experiments involving stimulation with ephrin-B2/Fc or EphB4/Fc fusion proteins, we show signals as normalized integrated density (Norm. IntDen) because the cell area is not significantly changed (Figure 8 and Figure 8—figure supplement 1B-D).

Finally, to gain further insight into the link between EphB4, *CAV1* and CD36, we have conducted an additional experiment, which revealed a very substantial and significant reduction of CD36 at the cell surface after *EPHB4* and *CAV1* knockdown (Figure 7D and E), whereas total CD36 remains unchanged (Figure 7B-E). Interestingly, CD36 in the Golgi, presumably reflecting newly synthesized CD36 and thereby compensatory upregulation of the fatty acid transporter, is increased in *siEPHB4* and *siCAV1* treated HUVECs (Figure 7—figure supplement 1B). As CD36 localization in the plasma membrane is critical for its transporter activity, the new findings provide a compelling explanation for the reduced uptake of BODIPY C12 and connect our data on EphB4, caveolae, Src signaling and *CAV1* phosphorylation to the metabolic alterations seen in vivo.

4) In Figure 8A, treatment of HUVEC cells with B2/Fc induced pCav1 but not total Cav1. This is inhibited by PP2 (inhibitor for Src signalling). Figure 8B, C shows quantifications making the point that B2/Fc stimulate both pCav1 and pCav1/Cav1, and that increases in both are inhibited in the presence of PP2. Figure 8C shows the PP2 effect on pCav1 in cells. Figure 8D show that the increase in pCav1/Cav1 with B2/Fc is not inhibited by Ly294002 and U0126 (inhibitors of PI3K and AKT). However, in Figure 8E, quantification of the increase in pCav1 in the cytoplasm versus golgi of HUVEC cells with B2/Fc, as judged by immunostaining, are shown in the presence of DMSO and inhibitors. Here, the PP2 inhibition is confirmed (PP2 also inhibits the basal level of pCav1 without B2/Fc stimulation. But LY294002 and U0126 also inhibit the B2/Fc-induced increase in pCav1, which seems to conflict with Figure 8D. There is apparently no impact on golgi localisation of pCav1. Please explain.

In line with the answer to the previous comment, we decided to show in Figure 8E the Norm.IntDen for the whole pCAV1 immunosignal and removed the data showing quantitation of the Golgi and cytoplasmic fractions. The revised versions of Figure 8C, E show ephrin-B2/Fc stimulation leads to a significant increase in pCAV1 signal, which is reduced to 51% by PP2 administration. As pointed out in your question, PP2 also lowers pCAV1 in Ctrl Fc stimulated cells (to 63%), which is consistent with the Western blot data (Figure 8A, B) and presumably reflects contributions by endogenous ephrin-B2/EphB4 interactions or other processes controlling Src activity. The AKT inhibitor LY294002 and ERK inhibitor U0126 have no significant effect on ephrin-B2/Fc-stimulated cells relative to vehicle control (DMSO; Figure 8E) even though the response seems a little blunted. While the cause of this might be indirect (e.g. through more general alterations in cell behavior), it is clear that only PP2 has a profound and significant effect on pCAV1 signals.